# K_2_CO_3_-Promoted Formal [3+3]-Cycloaddition of *N*-Unsubstituted Isatin *N*,*N′*-Cyclic Azomethine Imine 1,3-Dipoles with Knoevenagel Adducts

**DOI:** 10.3390/molecules28031034

**Published:** 2023-01-19

**Authors:** Guosheng Yang, Sicheng Li, Qiumi Wang, Huabao Chen, Chunping Yang, Zhongqiong Yin, Xu Song, Li Zhang, Cuifen Lu, Guizhou Yue

**Affiliations:** 1College of Science, Sichuan Agricultural University, Ya’an 625014, China; 2The Yingjing County Emergency Management Agency, Ya’an 625200, China; 3College of Agronomy, Sichuan Agricultural University, Chengdu 611130, China; 4College of Veterinary Medicine, Sichuan Agricultural University, Chengdu 611130, China; 5Hubei Collaborative Innovation Center for Advanced Organochemical Materials & Ministry-of-Education Key Laboratory for the Synthesis and Application of Organic Functional Molecules, Hubei University, Wuhan 430062, China

**Keywords:** azomethine ylide, cycloaddition, Knoevenagel adduct, spiropyridazine oxoindole, synthesis

## Abstract

The synthesis of dicyclic spiropyridazine oxoindole derivatives by using [3+3]-cycloaddition of *N*-unsubstituted isatin *N*,*N*′-cyclic azomethine imine 1,3-dipoles was reported. The products bearing two consecutive stereocenters, including spiroquaternary stereocenters in one ring structure, can be effectively obtained in moderate to excellent yields (20–93%) and low to moderate diastereoselectivities (1:9–10:1 dr). The synthesized compounds (>35 examples) were characterized by single-crystal XRD, FTIR, NMR, and mass spectral analysis.

## 1. Introduction

Heterocycles are common structural units that are widely found in natural products, pharmaceuticals, and agrochemicals [1]. In particular, dinitrogen-fused heterocycles are the core moieties in many biologically active compounds. The *N*,*N*-bicyclic pyridazinylpyrazolidinone skeletons are structurally interesting, and have been investigated as a herbicide CGA 271,312 (I) [2], camptothecin mimics (II) as a DNA topoisomerase (Top1) inhibitor [3], a potent drug to inhibit acetyl-CoA carboxylase (ACC) (III) [4], and β-sheet mimetic as a protease inhibitor (IV) [5] (Figure 1). Therefore, the exploration of practical and efficient methods for the synthesis of dinitrogen-fused heterocycles has attracted extensive attention in the field of organic chemistry and pharmacology. Various methods for the construction of *N*,*N*-bicyclic pyridazinylpyrazolidinone skeletons have been reported, including the double acylation of pyrazolidine [2]; the double alkylation of pyrazolidinone [4] and indazole [6]; the annulation by Rh-catalyzed C-H activation [7,8]; Rh-catalyzed [4+1] cyclization of *N*-arylphthalazine-1,4-dione with 2-diazo carbonyl compounds [9] or vinylene carbonate [10]; and the cycloaddition of 1,3-dipolar azomethine imines with various partners [11,12,13]. There is no doubt that the [3+3]-cycloaddition of azomethine imines with various dipolarophile precursors is one of the most popular methods in various protocols (Figure 1).

Very recently, Moghaddam and co-workers [14] reported an unexpected abnormal tandem Michael addition/*N*-cyclization of isatin *N*,*N*′-cyclic azomethine imine 1,3-dipole with 2-arylidenemalononitriles for the synthesis of spiropyridazine oxindoles with good yields (77–91%) and excellent diastereoselectivities (>20:1 dr), under the condition of DABCO as the base and DCM as the solvent at room temperature (Figure 1). However, the result of the reaction of *N*-unsubstituted isatin *N*,*N*′-cyclic azomethine imines with 2-arylidenemalononitriles was not exhibited under their optimal condition in this work. We found that the desired products of the reaction of *N*-unsubstituted isatin *N*,*N*′-cyclic azomethine imines with 2-benzylidenemalononitrile were not observed in the above optimal condition, even under reflux conditions (Table 1, Entry 1).

To date, the reactions of isatin *N*,*N*′-cyclic azomethine imines have rarely been studied and demonstrated by a few examples [15,16,17,18,19,20,21,22,23,24,25,26,27,28]. So, it is urgent to explore the new 1,3-dipolar cycloaddition of isatin *N*,*N*′-cyclic azomethine imines. Knoevenagel adducts, easily prepared from aromatic aldehydes and malononitrile in the presence of a base, were valuable substrates for various types of reactions such as Michael addition, cycloaddition reaction, reductive reaction, domino reaction, etc. [29,30,31,32,33,34,35,36,37]. Although the above work is excellent, it is still necessary to further expand the reaction range of istain *N*,*N*′-cyclic azomethine imines by using *N*-unsubstituted azomethine imines and 2-arylidenemalononitriles. Based on our previous studies of the reactions of azomethine imines and ylides 1,3-dipoles [38,39,40,41,42,43,44,45], herein we report the synthesis of spiropyridazinylpyrazolidinone oxoindole derivatives via the K_2_CO_3_-promoted formal [3+3]-cycloaddition of *N*-unsubstituted isatin *N*,*N*′-cyclic azomethine imines with Knoevenagel adducts under the condition of the DCE as the solvent at reflux.

## 2. Results and Discussion

At the beginning of our study, *N*-unsubstituted isatin *N*,*N*′-cyclic azomethine imine **1a** and 2-benzylidenemalononitrile **2a** were chosen as model substrates to search for the reaction optimal condition (Table 1). We discovered that isatin *N*,*N*′-cyclic azomethine imine **1a** could react with **2a** to give the desired product in 45% yield and with 2.5:1 dr value with K_2_CO_3_ as the base in dimethylsulfoxide (DMSO) in our previous work [43]. Inspired by the above result, we decided to develop an efficient method for the synthesis of *N*-unsubstituted spiropyridazinylpyrazolidinone oxoindoles as an important supplement to Moghaddam’s work. Initially, we screened various solvents in the presence of 2.0 equiv. K_2_CO_3_. Except for DMSO, other aprotic polar solvents (*N*,*N*-dimethylforammide (DMF), *N*,*N*-dimethylacetamide (DMA), pyrrolidone) could improve the product yields (47–76%) (Entries 3–5), while the products could be afforded in unsatisfied results with the yields of 32% and 42%, respectively, when *N*-methylpyrrolidone (NMP) and hydroxylethylpyrrolidone (HEP) were chosen as the reaction solvents. We also found that the results of protic solvents (EtOH, MeOH, and H_2_O) and aprotic solvents (THF, ACN, dioxane, and toluene) were inferior to those of DMSO. We turned our attention to chlorinated solvents and found that DCE used as the solvent could greatly enhance the product yield (77%) and diastereoselectivity (7.2:1 dr) (Entry 17). Next, we chose DCE as a solvent and tested a variety of bases to improve the diastereoselectivity of the reaction. The results demonstrated that the inorganic base Na_2_CO_3_ provided the product with 41% yield, but no product was formed with Li_2_CO_3_ and Cs_2_CO_3_ as the bases at reflux (Entries 18 and 20). Stronger bases NaOH, KOH, MeONa, and EtONa cannot improve the product yield (Entries 21–24). Additionally, the products could be acquired with lower yields or could not be observed when common organic bases (TEA, DIPEA, DBU, and DABCO) were used. According to the above experiments, K_2_CO_3_ was an optimal base. Then, we changed the amount of base, 2-benzylidenemalononitrile, and the concentration of the reaction. We found that the yield of the product did not change significantly, but diastereoselectivities were reduced when the amount of K_2_CO_3_ was changed compared with 2.0 equivalent K_2_CO_3_ (Entries 31–36, except for Entry 34). When the amount of 2-benzylidenemalononitrile **2a** was added up to 2.2–3.3 equivalent, all the yields were greatly increased, but the diastereoselectivities dropped off (Entries 37–39). Finally, the concentration of the reaction had little effect on yields and selectivities (Entries 40–41). The optimal reaction condition for 1,3-dipolar cycloaddition was established, and the desired product could be obtained in 77% yield and 7.2:1 dr value when using isatin *N*,*N*′-cyclic azomethine imine **1a** (1 equiv.), 2-benzylidenemalononitrile **2a** (1.1 equiv.), and K_2_CO_3_ (2.0 equiv.) as the solvent in DCE at reflux for 1.8 h (Entry 17).

After the optimal reaction condition was established, a wide scope of different substituted 2-benzylidenemalononitrile **2** was explored for this [3+3]-cycloaddition. As outlined in Table 2, various substituted groups on the phenyl ring of **2** could be tolerated, with the desired products afforded in low to excellent yields (26–89%) and low to moderate diastereoselectivities (1.6:1–10:1 dr) (Table 2, Entries 2–23). The reaction of **1a** with **2a** could be carried out at 1.0 mmol scale, with the product obtained in 86% yield with a 6.3:1 dr value under the optimal reaction condition.

It is worth mentioning that the progress of the reaction was monitored by TLC with visible light, UV, ninhydrin, and the color change of the reaction solution (Figure 2). Substrate **2** bearing electron-donating or electron-withdrawing groups led to inferior results in contrast with **2a**. The position of substituents on the benzene ring had a great impact on the yields, and the 2-substituted phenyl ring of **2a** usually gave the best yields (Entries 2, 5, 8, and 11), except for 2-BrPh (Entry 14). It was regrettable that isomers **3** and **3′** were not separated by column chromatography. Fortunately, in most reactions, a single isomer with sufficient purity can be obtained by using EtOH recrystallization, which can be characterized by different spectroscopic techniques, such as FTIR, ^1^H and ^13^C NMR, and mass spectrometry (see Appendix A).

Subsequently, various substituents on the benzene ring of isatin *N*,*N*′-cyclic azomethine imines can also be well tolerated, which further proves the universality of the cycloaddition reaction. Unfortunately, the diastereoselectivities (3.3:1–1:2 dr) of the desired product with electron-donating groups (5-Me, 5-OMe) or electron-withdrawing groups (5-F, 5-Cl, 6-Br, 5-I, 5-NO_2_, and 7-CF_3_) on the benzene ring of the compound **1** was lower than that of the model reaction, as shown in Table 3. The reaction of 5-methyl isatin *N*,*N*′-cyclic azomethine imine was scaled up to 5.0 mmol, and the products **4a**/**4′a** could be obtained in 75% yield. The structure of **4a** was clearly confirmed by single-crystal X-ray diffraction (Figure 3) [46]. The strong electron-drawing group NO_2_ significantly impacted the product yield (Entry 7). The **4g**/**4′g** was only obtained in 20% yield with 2.5:1 dr under the optimal reaction condition (Table 1, Entry 18) and obtained in 40% yield with 1:1.4 dr under the suboptimal condition (Table 1, Entry 38), while 7-trifluoromethylisatin *N*,*N*′-cyclic azomethine imine led to a satisfying yield of 74% with 1.6:1 dr (Entry 8). To our delight, the single isomers **4** could also be obtained through the recrystallization from EtOH in all reactions except for **4a**/**4′a** and **4h**/**4′h**.

To expand the application scope of the reaction, α-methyl isatin *N*,*N′*-cyclic azomethine imines **1** reacted with **2** in the standard condition to afford cycloadducts **5**/**5′** in 58–93% yields with 1.3:1–2.8:1 dr under the optimal condition (Table 4). However, only a trace amount of desired products could be observed when the R^2^ group of isatin *N*,*N′*-cyclic azomethine imines was replaced by methyl or phenyl (Entries 8 and 9). In most reactions, the single isomers **5a**, **5d**, **5e**, and **5f** were also afforded by recrystallization from EtOH.

The dicyclic spiropyridazine oxoindole product exhibited potentially wide application in organic synthesis (Figure 2). For example, the *N*-allylation of **3a** reacted with Morita–Baylis–Hillman carbonate to provide the compound **6** in 76% yield in the presence of DABCO. **3a** reacted with Boc_2_O to give the *N*,*N*,*N*-triBoc-protected product **7** in 71% yield. The diazotization and hydrolysis of the amino group of **3′b** could generate the enol **8** in 25% yield under NaNO_2_/*p*-TsOH [47]. Finally, **5a** can be converted into *N*-acetyl product **9** with a high yield (86%).

On the basis of the literature reports [14], experimental results, and X-ray analysis, a plausible reaction pathway is proposed in Figure 3. The intermediate **I** could be obtained by resonance form of the compound **1**. The intermediate **I** was tautomerized under the condition of the base to form the delocalized intermediate **II**, which can be added with Knoevenagel adducts **2** through Michael addition to give the intermediate **III**. Subsequently, the nucleophilic addition of the nitrogen atom of the intermediate **III** to the cyano group could give the formal [3+3]-cycloadduct **IV**. The final products **3**/**3′, 4**/**4′**, and **5**/**5′** were formed via the enamination of the imine group of the compound **IV**.

## 3. Materials and Methods

### 3.1. General Methods

All reactions were carried out without strict water-free and oxygen-free conditions, except for using NaH, MeONa, and EtONa as bases. All reagents and reagents were obtained from commercial suppliers and were directly used for reactions without further purification unless otherwise stated. When the reactions preformed at the condition of NaH, MeONa, and EtONa, solvent DCE was predried with CaH_2_. Flash chromatography was performed using silica gel (200−300 mesh). Reactions were monitored by TLC or/and color changes of the reaction solution. Visualization was achieved under a UV lamp (254 nm and 365 nm), I_2_, and by developing the plates with ninhydrin. ^1^H and ^13^C NMR were recorded on 400 and 600 MHz NMR spectrometers with tetramethylsilane (TMS) as the internal standard. IR spectra were acquired on an FTIR spectrometer and were reported in wavenumbers (cm^−1^). High-resolution mass spectra were obtained using electrospray ionization (ESI). ^1^H NMR splitting patterns are designated as singlet (s), broad singlet (brs), double (d), triplet (t), false triplet (ψt), quartet (q), doublet of doublets (dd), multiples (m), etc. Coupling constants (*J*) are reported in Hertz (Hz).

### 3.2. Preparation of Intermediates

Pyrazolidine-3-ones were obtained by the reaction of hydrazine monohydrate with methyl acrylate in ethanol under the refluxing condition [15]. All isatin *N*,*N′*-cyclic azomethine imines **1** were prepared by the condensation of isatins and the above pyrazolidone in menthol under 45 °C or the refluxing condition [15]. All Knoevenagel adducts **2** were prepared by a one-step reaction (0.2 equiv. KOH, 1.0 equiv. aldehyde, 1.0 equiv. malononitrile, and EtOH/H_2_O at room temperature for 2–3 h) [48], except for **2w** (0.1 equiv. DABCO and MeOH).

### 3.3. General Procedure for Condition Optimization

A 10.0 mL tube was charged with isatin *N*,*N′*-cyclic azomethine imine **1a** (0.5 mmol, 1.0 equiv.), Knoevenagel adduct **2a** (0.55–1.65 mmol, 1.1–3.3 equiv.), base (0.5–2.0 mmol, 0.5–4.0 equiv.), and solvent (1.0–4.0 mL). The suspended solution was vigorously stirred at rt or reflux, and then the base was added. The reaction finished when the suspension reaction liquid gradually changed from red to brown and then to green. The solution was purified by flash silica gel chromatography eluted with EtOAc-petroleum ether (1:3 to 1:0) to afford the corresponding products **3a** and **3′a**.

### 3.4. General Procedure for Typical Procedure for [3+3]-Cycloaddition

A tube (25.0 mL) was charged with isatin *N*,*N′*-cyclic azomethine imine **1a** (1.0 mmol, 1.0 equiv.), Knoevenagel adduct **2** (1.1 mmol, 1.1 equiv.), K_2_CO_3_ (276.4 mg, 2.0 mmol, 2.0 equiv.), and DCE (4.0 mL). The suspended solution was vigorously stirred at reflux. When the reaction mixture became clear and the color of the reaction solution changed, the reaction finished (1–18 h). The solution was purified by flash silica gel chromatography eluted with EtOAc-petroleum ether (1:3 to 1:0) to afford the corresponding cycloaddition products **3**/**3′**, **4**/**4′**, and **5**/**5′**.

### 3.5. Synthesis of the Products 6, 7, 8, and 9

DABCO (2.2 mg, 0.2 mmol, 0.1 equiv.) was added to a solution of **3a** (43 mg, 0.2 mmol, 1.0 equiv.) and ethyl 2-(((tert-butoxycarbonyl)oxy)methyl)acrylate (92 mg, 0.4 mmol, 2.0 equiv.) in DCM (3.0 mL) at rt. The mixture was stirred at rt for 5.0 h. The resulting mixture was washed with saturated NH_4_Cl solution (10.0 mL). The aqueous solution was extracted with EtOAc (3 × 10.0 mL). The combined organic layers were dried and concentrated. The crude product was purified by flash column chromatography on silica gel with petroleum ether-EtOAc (3:1 to 1:1) to obtain the compound **6**.

Boc_2_O (175 mg, 0.8 mmol, 4.0 equiv.) was added to a solution of **3a** (74 mg, 0.2 mmol, 1.0 equiv.) and DMAP (5.0 mg, 0.04 mmol, 0.2 equiv.) in DCM (3.0 mL) at 0 °C. The mixture was stirred at rt for 5.0 h. The resulting mixture was washed with saturated NH_4_Cl solution (10 mL). The aqueous solution was extracted with EtOAc (3 × 10 mL). The combined organic layers were dried and concentrated. The crude product was purified by flash column chromatography on silica gel with petroleum ether-EtOAc (3:1 to 1:1) to give the adduct **7**.

To a solution of *p*-TsOH·H_2_O (114 mg, 0.6 mmol) in MeCN (1.5 mL) was added with **3′b** (76 mg, 0.2 mmol. 1.0 equiv.). The resulting suspension of amine salt was cooled to 5 °C and was gradually added with a solution of NaNO_2_ (28 mg, 0.4 mmol) in H_2_O (0.2 mL). The reaction mixture was stirred for 10 min, then allowed to warm to rt. The mixture was stirred at rt for 3.0 h. The reaction mixture was then added with H_2_O (2.0 mL), NaHCO_3_ (1 M; until pH = 9–10), and Na_2_S_2_O_3_ (2 M, 1.0 mL). The aqueous solution was extracted with EtOAc (3 × 15.0 mL). The combined organic layers were dried and concentrated. The crude product was purified by flash column chromatography on silica gel with petroleum ether-EtOAc (2:1 to 1:2) to form the compound **8**.

Ac_2_O (41 mg, 0.4 mmol, 2.0 equiv.) was added to a solution of **5a** (76 mg, 0.2 mmol, 1.0 equiv.) and DMAP (5.0 mg, 0.04 mmol, 0.2 equiv) in DCM (3.0 mL) at 0 °C. The mixture was stirred at rt for 5.0 h. The resulting mixture was washed with saturated NH_4_Cl solution (10 mL). The aqueous solution was extracted with EtOAc (3 × 15.0 mL). The combined organic layers were dried and concentrated. The crude product was purified by flash column chromatography on silica gel with petroleum ether-EtOAc (3:1 to 1:1) to afford the compound **9**.

### 3.6. Data for All New Compounds

(±)-(3*R*,6′*R*)-8′-amino-1′,2-dioxo-6′-phenyl-1′*H*,6′*H*-spiro[indoline-3,5′-pyrazolo[1,2-a]pyridazine]-7′-carbonitrile (**3a**)



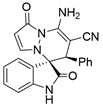



[Reaction time: 4.0 h], 318 mg, 86%, a white solid, m.p. 214.5–216.5 °C; IR (thin film): ν_max_ 3367, 3275, 3147, 3062, 3026, 2183, 1712, 1684, 1625, 1572, 1432, 1285, 1048, 752 cm^−1^; ^1^H NMR (400 MHz, DMSO-*d*_6_): δ 10.40 (s, 1 H), 7.74–7.71 (m, 3 H), 7.32 (d, *J* = 4.0 Hz, 1 H), 7.30 (td, *J* = 7.6, 0.8 Hz, 1 H), 7.18–7.11 (m, 4 H), 6.97 (brs, 2 H), 6.61 (d, *J* = 7.6 Hz, 1 H), 5.67 (d, *J* = 4.0 Hz, 1 H), 4.64 (s, 1 H); ^13^C NMR (100 MHz, DMSO-*d*_6_): δ 171.1, 165.4, 150.2, 142.8, 142.6, 142.6, 134.7, 131.9, 130.1, 128.4, 128.2, 126.0, 123.1, 122.9, 119.6, 119.6, 110.6, 101.0, 67.4, 58.4, 46.6; HRMS (ESI): *m*/*z* calcd. for C_21_H_15_N_5_O_2_ [M + H]^+^ 370.1304 found 370.1380.

(±)-(3*R*,6′*S*)-8′-amino-1′,2-dioxo-6′-(*o*-tolyl)-1′*H*,6′*H*-spiro[indoline-3,5′-pyrazolo[1,2-a]pyridazine]-7′-carbonitrile (**3′b**)



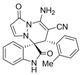



[Reaction time: 2.0 h], 249 mg, 65%, a white solid, m.p. 210.3–212.1 °C; IR (thin film): ν_max_ 3369, 3282, 3146, 3116, 3026, 2189, 1724, 1662, 1530, 1331, 1159, 1047, 758 cm^−1^; ^1^H NMR (400 MHz, DMSO-*d*_6_): δ 10.88 (s, 1 H), 7.73 (brs, 2 H), 7.39(d, *J* = 4.0 Hz, 1 H), 7.31 (td, *J* = 7.6, 0.8 Hz, 1 H), 7.21–7.10 (m, 2 H), 7.03 (d, *J* = 6.8 Hz, 1 H), 6.96 (d, *J* = 7.6 Hz, 1 H), 6.87 (d, *J* = 8.0 Hz, 1 H), 6.76 (ψt, *J* = 7.6 Hz, 1 H), 5.83 (d, *J* = 7.6 Hz, 1 H), 5.62 (d, *J* = 4.0 Hz, 1 H), 4.28 (s, 1 H), 1.71 (s, 3 H); ^13^C NMR (100 MHz, DMSO-*d*_6_): δ 172.1, 165.2, 149.8, 142.7, 142.4, 138.1, 135.7, 131.6, 130.4, 128.7, 128.4, 126.4, 126.0, 122.2, 121.7, 120.0, 111.0, 100.2, 66.6, 58.3, 40.9, 19.2; HRMS (ESI): *m*/*z* calcd. forC_22_H_17_N_5_O_2_ [M + H]^+^ 384.1460, found 384.1468. 

(±)-(3*R*,6′*S*)-8′-amino-1′,2-dioxo-6′-(*m*-tolyl)-1′*H*,6′*H*-spiro[indoline-3,5′-pyrazolo[1,2-a]pyridazine]-7′-carbonitrile (**3′c**)



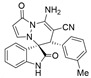



[Reaction time: 4.0 h], 228 mg, 59%,a white solid, m.p. 211.1–212.9 °C; IR (thin film): *ν*_max_ 3481, 3321, 3252, 3147, 2185, 1735, 1686, 1604, 1560, 1466, 1426, 1286, 1200, 1175, 762 cm^−1^; ^1^H NMR (400 MHz, DMSO-*d*_6_): δ 10.86 (s, 1 H), 7.77–7.76 (m, 2 H), 7.62 (d, *J* = 4.0 Hz, 1 H), 7.24 (ψt, *J* = 7.6 Hz, 1 H), 7.06–7.02 (m, 2 H), 6.91 (ψt, *J* = 7.6 Hz, 1 H), 6.73 (d, *J* = 8.0 Hz, 1 H), 6.68 (brs, 2 H), 6.53 (d, *J* = 7.6 Hz, 1 H), 5.64 (d, *J* = 4.0 Hz, 1 H), 4.26 (s, 1 H), 2.13 (s, 3 H); ^13^C NMR (100 MHz, DMSO-*d*_6_): δ 171.1, 165.4, 150.1, 142.8, 142.6, 137.1, 134.7, 131.8, 130.8, 129.1, 128.1, 127.2, 126.0, 123.1, 122.9, 119.6, 110.6, 100.9, 67.4, 58.6, 46.5, 21.4; HRMS (ESI): *m*/*z* calcd. for C_22_H_17_N_5_O_2_ [M + H]^+^ 384.1460, found 384.1464.

(±)-(3*R*,6′*R*)-8′-amino-1′,2-dioxo-6′-(*p*-tolyl)-1′*H*,6′*H*-spiro[indoline-3,5′-pyrazolo[1,2-a]pyridazine]-7′-carbonitrile (**3d**)



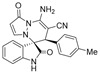



[Reaction time: 4.0 h], 230 mg, 60%, a yellow solid, m.p. 210.5–212.5 °C; IR (thin film): ν_max_ 3440, 3363, 3230, 3146, 2197, 1724, 1688, 1613, 1570, 1542, 1518, 1468, 1257, 1194, 752 cm^−1^; ^1^H NMR (400 MHz, DMSO-*d*_6_): δ 10.39 (s, 1 H), 7.72 (d, *J* = 7.2 Hz, 1 H), 7.68 (s, 2 H), 7.32–7.28 (m, 2 H), 7.16 (ψt, *J* = 7.4 Hz, 1 H), 6.94 (d, *J* = 7.6 Hz, 2 H), 6.85 (*br* s, 2 H), 6.62 (d, *J* = 8.0 Hz, 1 H), 5.66 (d, *J* = 4.0 Hz, 1 H), 4.61 (s, 1 H), 2.18 (s, 3 H); ^13^C NMR (100 MHz, DMSO-*d*_6_): δ 171.1, 165.4, 150.1, 142.8, 142.5, 137.5, 131.8, 131.7, 130.0, 128.9, 126.0, 123.1, 123.0, 119.6, 110.7, 100.9, 67.4, 58.7, 46.2, 21.1; HRMS (ESI): *m*/*z* calcd. for C_22_H_17_N_5_O_2_ [M + H]^+^ 384.1460, found 384.1461.

(±)-(3*R*,6′*R*)-8′-amino-1′,2-dioxo-6′-(2-methoxyphenyl)-1′*H*,6′*H*-spiro[indoline-3,5′-pyrazolo[1,2-a]pyridazine]-7′-carbonitrile (**3e**)



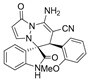



[Reaction time: 2.0 h], 275 mg, 69%, a white solid, m.p. 238.9–240.2 °C; IR (thin film): ν_max_ 3369, 3270, 3136, 2186, 1714, 1677, 1620, 1569, 1509, 1471, 1426, 1253, 1189, 744 cm^−1^; ^1^H NMR (400 MHz, DMSO-*d*_6_): δ 10.39 (s, 1 H), 7.71 (d, *J* = 7.2 Hz, 1 H), 7.67 (s, 2 H), 7.31 (ψt, *J* = 7.6 Hz, 1 H), 7.30 (d, *J* = 4.0 Hz, 1 H), 7.16 (ψt, *J* = 7.6 Hz, 1 H), 6.87 (brs, 2 H), 6.69 (d, *J* = 8.0 Hz, 2 H), 6.63 (d, *J* = 7.6 Hz, 1 H), 5.66 (d, *J* = 4.0 Hz, 1 H), 4.60 (s, 1 H), 3.66 (s, 3 H); ^13^C NMR (100 MHz, DMSO-*d*_6_): δ 171.2, 165.4, 159.2, 150.0, 142.9, 142.5, 131.2, 126.4, 126.0, 123.1, 123.0, 119.6, 113.6, 100.6, 100.9, 67.4, 58.9, 55.4, 45.8; HRMS (ESI): *m*/*z* calcd. for C_22_H_17_N_5_O_3_ [M + H]^+^ 400.1410, found 400.1405.

(±)-(3*R*,6′*R*)-8′-amino-1′,2-dioxo-6′-(3-methoxyphenyl)-1′*H*,6′*H*-spiro[indoline-3,5′-pyrazolo[1,2-a]pyridazine]-7′-carbonitrile (**3f**)



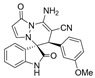



[Reaction time: 7.0 h], 171 mg, 43%, a white solid, m.p. 242.5–243.9 °C; IR (thin film): ν_max_ 3440, 3363, 3230, 3146, 2197, 1724, 1688, 1613, 1570, 1542, 1518, 1468, 1257, 1194, 752 cm^−1^; ^1^H NMR (400 MHz, DMSO-*d*_6_): δ 10.45 (s, 1 H), 7.73 (d, *J* = 8.0 Hz, 1 H), 7.72 (s, 2 H), 7.34 (d, *J* = 4.0 Hz, 1 H), 7.32 (ψt, *J* = 7.6 Hz, 1 H), 7.18 (ψt, *J* = 7.6 Hz, 1 H), 7.06 (ψt, *J* = 7.6 Hz, 1 H), 6.73 (dd, *J* = 8.0, 2.0 Hz, 1 H), 6.63 (d, *J* = 7.6 Hz, 1 H), 6.53 (brs, 2 H), 5.67 (d, *J* = 4.0 Hz, 1 H), 4.64 (s, 1 H), 3.55 (s, 3 H); ^13^C NMR (100 MHz, DMSO-*d*_6_): δ 171.1, 165.4, 158.8, 150.1, 142.9, 142.6, 136.2, 131.8, 129.2, 126.0, 123.1, 122.9, 122.3, 119.6, 115.8, 113.8, 110.6, 101.0, 67.3, 58.4, 55.2, 46.5; HRMS (ESI): *m*/*z* calcd. for C_22_H_17_N_5_O_3_ [M + H]^+^ 400.1410, found 400.1410.

(±)-(3*R*,6′*R*)-8′-amino-1′,2-dioxo-6′-(4-methoxyphenyl)-1′*H*,6′*H*-spiro[indoline-3,5′-pyrazolo[1,2-a]pyridazine]-7′-carbonitrile (**3g**)



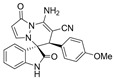



[Reaction time: 2.0 h], 263 mg, 66%, a yellow solid, m.p. 245.2–247.1 °C; IR (thin film): ν_max_ 3684, 3987, 1710, 1694, 1573, 1383, 1267, 1128, 761 cm^−1^; ^1^H NMR (400 MHz, DMSO-*d*_6_): δ 10.39 (s, 1 H), 7.71 (d, *J* = 7.6 Hz, 1 H), 7.67 (s, 2 H), 7.31 (ψt, *J* = 8.0 Hz, 1 H), 7.30 (d, *J* = 4.0 Hz, 1 H), 7.16 (ψt, *J* = 7.6 Hz, 1 H), 6.87 (brs, 2 H), 6.69 (d, *J* = 8.0 Hz, 2 H), 6.63 (d, *J* = 7.6 Hz, 1 H), 5.66 (d, *J* = 4.0 Hz, 1 H), 4.60 (s, 1 H), 3.65 (s, 3 H); ^13^C NMR (100 MHz, DMSO-*d*_6_): δ 171.2, 165.4, 159.2, 150.0, 142.9, 142.6, 131.8, 131.2, 126.4, 126.0, 123.1, 123.0, 119.7, 113.6, 110.7, 100.9, 67.4, 58.8, 55.4, 45.7; HRMS (ESI): *m*/*z* calcd. for C_22_H_17_N_5_O_3_ [M + H]^+^ 400.1410, found 400.1419.

(±)-(3*R*,6′*R*)-8′-amino-1′,2-dioxo-6′-(2-fluorophenyl)-1′*H*,6′*H*-spiro[indoline-3,5′-pyrazolo[1,2-a]pyridazine]-7′-carbonitrile (**3h**)



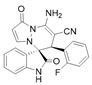



[Reaction time: 1.0 h], 302 mg, 78%, a yellow solid, m.p. 192.7–193.2 °C; IR (thin film): ν_max_ 3479, 3313, 2188, 1734, 1687, 1618, 1563, 1471, 1428, 1333, 1192, 1097, 758 cm^−1^; ^1^H NMR (400 MHz, DMSO-*d*_6_): δ 10.60 (s, 1 H), 7.78 (s, 2 H), 7.65 (d, *J* = 7.6 Hz, 1 H), 7.35 (td, *J* = 7.6, 1.6 Hz, 1 H), 7.33 (d, *J* = 4.0 Hz, 1 H), 7.31 (td, *J* = 7.6, 0.8 Hz, 1 H), 7.27–7.21 (m, 1 H), 7.17 (td, *J* = 7.6, 0.8 Hz, 1 H), 7.13 (ψt, *J* = 7.6 Hz, 1 H), 6.91 (ψt, *J* = 9.2 Hz, 1 H), 6.66 (d, *J* = 7.6 Hz, 1 H), 5.67 (d, *J* = 4.0 Hz, 1 H), 4.98 (s, 1 H); ^13^C NMR (100 MHz, DMSO-*d*_6_): δ 171.0, 165.4, 150.5, 142.6, 142.4, 132.0, 131.2, 130.6 (d), 126.3, 124.8, 124.7, 123.1, 122.5, 121.9 (d), 119.4, 115.4 (d), 110.6, 100.9, 66.9, 57.5, 38.1; ^19^F NMR (376 MHz, DMSO-*d*_6_): δ −114.5; HRMS (ESI): *m*/*z* calcd. for C_21_H_14_FN_5_O_2_ [M + H]^+^ 388.1210, found 388.1216.

(±)-(3*R*,6′*R*)-8′-amino-1′,2-dioxo-6′-(2-fluorophenyl)-1′*H*,6′*H*-spiro[indoline-3,5′-pyrazolo[1,2-a]pyridazine]-7′-carbonitrile (**3i**)



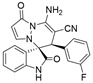



[Reaction time: 2.0 h], 256 mg, 66%, a yellow solid, m.p. 246.3–248.1 °C; IR (thin film): ν_max_ 3365, 3269, 3147, 2184, 1717, 1682, 1623, 1572, 1550, 1476, 1431, 1269, 1194, 893, 748 cm^−1^; ^1^H NMR (400 MHz, DMSO-*d*_6_): δ 10.52 (s, 1 H), 7.76 (s, 2 H), 7.73 (d, *J* = 7.6 Hz, 1 H), 7.36 (d, *J* = 4.0 Hz, 1 H), 7.33 (td, *J* = 7.6, 0.8 Hz, 1 H), 7.18 (ψt, *J* = 7.4 Hz, 2 H), 7.03 (td, *J* = 8.4, 2.4 Hz, 1 H), 6.78 (brs, 2 H), 6.65 (d, *J* = 7.6 Hz, 1 H), 5.69 (d, *J* = 4.0 Hz, 1 H), 4.73 (s, 1 H); ^13^C NMR (100 MHz, DMSO-*d*_6_): δ 171.0, 165.4, 150.3, 142.8, 142.7, 137.8 (d), 132.0, 130.2 (d), 126.3, 126.0, 123.3, 122.6, 119.5, 116.7 (d), 115.5, 115.3, 110.8, 101.2, 66.7, 57.8, 46.1; ^19^F NMR (376 MHz, DMSO-*d*_6_): δ −113.5; HRMS (ESI): *m*/*z* calcd. for C_21_H_14_FN_5_O_2_ [M + H]^+^ 388.1210, found 388.1219.

(±)-(3*R*,6′*R*)-8′-amino-1′,2-dioxo-6′-(2-fluorophenyl)-1′*H*,6′*H*-spiro[indoline-3,5′-pyrazolo[1,2-a]pyridazine]-7′-carbonitrile (**3j**)



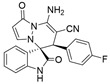



[Reaction time: 3.0 h], 225 mg, 58%, a yellow solid, m.p. 249.6–250.2 °C; IR (thin film): *ν*_max_ 3375, 3271, 3146, 2184, 1716, 1678, 1623, 1572, 1541, 1507, 1474, 1278, 1068, 907, 742 cm^−^; ^1^H NMR (400 MHz, DMSO-*d*_6_): δ 10.89 (s, 1 H), 7.81 (s, 2 H), 7.64 (d, *J* = 4.0 Hz, 1 H), 7.25 (td, *J* = 7.6, 0.8 Hz, 1 H), 7.02 (ψt, *J* = 8.8 Hz, 1 H), 6.94 (ψt, *J* = 7.6 Hz, 3 H), 6.75 (d, *J* = 8.0 Hz, 1 H), 6.53 (d, *J* = 7.6 Hz, 1 H), 5.65 (d, *J* = 4.0 Hz, 1 H), 4.36 (s, 1 H); ^13^C NMR (100 MHz, DMSO-*d*_6_): δ 171.0, 165.4, 150.2, 142.8, 142.7, 132.1, 132.0, 131.0 (d), 126.0, 123.2, 122.7, 119.6, 115.3, 115.1, 110.7, 101.0, 67.3, 58.2, 45.8; ^19^F NMR (376 MHz, DMSO-*d*_6_): δ −119.6; HRMS (ESI): *m*/*z* calcd. for C_21_H_14_FN_5_O_2_ [M + H]^+^ 388.1210, found 388.1217.

(±)-(3*R*,6′*S*)-8′-amino-6′-(2-chlorophenyl)-1′,2-dioxo-1′*H*,6′*H*-spiro[indoline-3,5′-pyrazolo[1,2-a]pyridazine]-7′-carbonitrile (**3′k**)



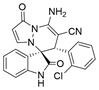



[Reaction time:3.0 h], 360 mg, 89%, a white solid, m.p. 240.9–241.6 °C; IR (thin film): ν_max_ 3402, 3271, 3136, 2183, 1736, 1683, 1626, 1575, 1540, 1473, 1257, 1042, 853, 752 cm^−1^; ^1^H NMR (400 MHz, DMSO-*d*_6_): δ 10.95 (s, 1 H), 7.85 (s, 2 H), 7.40–7.27 (m, 6 H), 6.90 (d, *J* = 8.0 Hz, 1 H), 6.74 (ψt, *J* = 7.4 Hz, 1 H), 5.65 (d, *J* = 4.0 Hz, 1 H), 4.40 (s, 1 H); ^13^C NMR (100 MHz, DMSO-*d*_6_): δ 172.0, 165.3, 150.0, 143.0, 142.5, 135.3, 131.8, 130.6, 130.4, 129.5, 128.0, 127.7, 125.8, 122.4, 121.0, 120.0, 111.2, 100.6, 65.9, 56.6, 41.7; HRMS (ESI): *m*/*z* calcd. for C_21_H_14_ClN_5_O_2_ [M + H]^+^ 404.0914, found 404.0914. 

(±)-(3*R*,6′*R*)-8′-amino-1′,2-dioxo-6′-(2-chlorophenyl)-1′*H*,6′*H*-spiro[indoline-3,5′-pyrazolo[1,2-a]pyridazine]-7′-carbonitrile (**3l**)



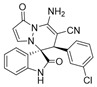



[Reaction time:2.0 h], 239 mg, 59%, a yellow solid, m.p. 243.6–244.8 °C; IR (thin film): ν_max_ 3488, 3324, 3243, 2186, 1734, 1687, 1614, 1565, 1469, 1428, 1333, 1197, 753, 697 cm^−1^; ^1^H NMR (400 MHz, DMSO-*d*_6_): δ 10.54 (s, 1 H), 7.77 (s, 2 H), 7.72 (d, *J* = 7.2 Hz, 1 H), 7.36 (d, *J* = 4.0 Hz, 1 H), 7.33 (td, *J* = 7.6, 1.2 Hz, 1 H), 7.27–7.24 (m, 1 H), 7.19 (td, *J* = 7.6, 0.8 Hz, 1 H), 7.01 (brs, 1 H), 6.90 (brs, 1 H), 6.65 (d, *J* = 8.0 Hz, 1 H), 5.69 (d, *J* = 4.0 Hz, 1 H), 4.73 (s, 1 H); ^13^C NMR (100 MHz, DMSO-*d*_6_): δ 170.9, 165.4, 150.3, 142.9, 142.7, 137.4, 132.8, 132.0, 130.1, 129.9, 128.9, 128.5, 126.1, 123.3, 122.6, 119.5, 110.8, 101.2, 67.2, 57.7, 46.1; HRMS (ESI): *m*/*z* calcd. for C_21_H_14_ClN_5_O_2_ [M + H]^+^ 404.0914, found 404.0918.

(±)-(3*R*,6′*R*)-8′-amino-1′,2-dioxo-6′-(2-chlorophenyl)-1′*H*,6′*H*-spiro[indoline-3,5′-pyrazolo[1,2-a]pyridazine]-7′-carbonitrile (**3m**)



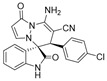



[Reaction time: 4.0 h], 219 mg, 54%, a yellow solid, m.p. 259.7–261.7 °C; IR (thin film): ν_max_ 3452, 3240, 3136, 2192, 1725, 1685, 1613, 1570, 1536, 1489, 1465, 1282, 1206, 838, 759 cm^−1^; ^1^H NMR (400 MHz, DMSO-*d*_6_): δ 10.49 (s, 1 H), 7.77 (s, 2 H), 7.73 (d, *J* = 7.6 Hz, 1 H), 7.36 (d, *J* = 4.0 Hz, 1 H), 7.33 (ψt, *J* = 7.6 Hz, 1 H), 7.25 (d, *J* = 7.6 Hz, 2 H), 7.18 (ψt, *J* = 7.6 Hz, 1 H), 6.97 (brs, 2 H), 6.66 (d, *J* = 8.0 Hz, 1 H), 5.69 (d, *J* = 4.0 Hz, 1 H), 4.72 (s, 1 H); ^13^C NMR (100 MHz, DMSO-*d*_6_): δ 170.9, 165.4, 150.2, 142.8, 142.7, 133.9, 133.1, 132.0, 131.9, 128.4, 126.0, 123.3, 122.6, 119.6, 110.8, 101.1, 67.2, 57.9, 45.9; HRMS (ESI): *m*/*z* calcd. for C_21_H_14_ClN_5_O_2_ [M + H]^+^ 404.0914, found. 404.0915.

(±)-(3*R*,6′*S*)-8′-amino-1′,2-dioxo-6′-(2-bromophenyl)-1′*H*,6′*H*-spiro[indoline-3,5′-pyrazolo[1,2-a]pyridazine]-7′-carbonitrile (**3′n**)



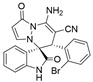



[Reaction time: 3.0 h], 265 mg, 59%, a yellow solid, m.p. 259.7–261.2 °C; IR (thin film): ν_max_ 3365, 3143, 2194, 1725, 1659, 1580, 1527, 1470, 1426, 1216, 1050, 745, 587 cm^−1^; ^1^H NMR (400 MHz, DMSO-*d*_6_): δ 10.96 (s, 1 H), 7.85 (s, 2 H), 7.49–7.44 (m, 2 H), 7.35–7.27 (m, 4 H), 6.91 (d, *J* = 8.0 Hz, 1 H), 6.73 (ψt, *J* = 7.6 Hz, 1 H), 5.66 (d, *J* = 4.0 Hz, 1 H), 5.64 (d, *J* = 8.8 Hz, 1 H), 4.36 (s, 1 H); ^13^C NMR (100 MHz, DMSO-*d*_6_): δ 172.0, 165.3, 150.0, 143.1, 142.4, 137.0, 132.7, 131.9, 130.7 (2C), 128.6, 126.9, 126.0, 122.4, 120.8, 120.0, 111.2, 100.5, 65.8, 56.7, 44.4; HRMS (ESI): *m*/*z* calcd. for C_21_H_14_BrN_5_O_2_ [M + H]^+^ 448.0409, found 448.0430. 

(±)-(3*R*,6′*R*)-8′-amino-1′,2-dioxo-6′-(3-bromophenyl)-1′*H*,6′*H*-spiro[indoline-3,5′-pyrazolo[1,2-a]pyridazine]-7′-carbonitrile (**3o**)



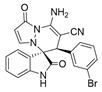



[Reaction time: 1.0 h], 319 mg, 71%, a yellow solid, m.p. 259.9–261.6 °C; IR (thin film): ν_max_ 3362, 3268, 3146, 2182, 1718, 1682, 1624, 1570, 1542, 1473, 1281, 1189, 749, 591 cm^−1^; ^1^H NMR (400 MHz, DMSO-*d*_6_): δ 10.54 (s, 1 H), 7.77 (s, 2 H), 7.72 (d, *J* = 7.6 Hz, 1 H), 7.40–7.32 (m, 3 H), 7.21–7.12 (m, 3 H), 6.94 (brs, 1 H), 6.66 (d, *J* = 7.6 Hz, 1 H), 5.69 (d, *J* = 4.0 Hz, 1 H), 4.71 (s, 1 H); ^13^C NMR (100 MHz, DMSO-*d*_6_): δ 170.9, 165.4, 150.3, 142.9, 142.7, 137.6, 132.8, 132.0, 131.4, 130.4, 129.2, 126.1, 123.3, 122.6, 121.4, 119.5, 110.8, 101.8, 101.2, 67.2, 57.7, 46.0; HRMS (ESI): *m*/*z* calcd. for C_21_H_14_BrN_5_O_2_ [M + H]^+^ 448.0409, found 448.0410.

(±)-(3*R*,6′*S*)-8′-amino-1′,2-dioxo-6′-(4-bromophenyl)-1′*H*,6′*H*-spiro[indoline-3,5′-pyrazolo[1,2-a]pyridazine]-7′-carbonitrile (**3′p**)



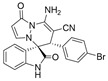



[Reaction time: 1.5 h], 319 mg, 71%, a yellow solid, m.p. 260.5–262.5 °C; IR (thin film): ν_max_ 3361, 3269, 2189, 1720, 1681, 1626, 1570, 1535, 1476, 1279, 1105, 750, 608 cm^−1^; ^1^H NMR (400 MHz, DMSO-*d*_6_): δ 10.96 (s, 1 H), 7.85 (s, 2 H), 7.49–7.44 (m, 2 H), 7.35–7.27 (m, 4 H), 6.91 (d, *J* = 7.6 Hz, 1 H), 6.73 (ψt, *J* = 7.6 Hz, 1 H), 5.66 (d, *J* = 4.0 Hz, 1 H), 5.64 (d, *J* = 7.6 Hz, 1 H), 4.37 (s, 1 H); ^13^C NMR (100 MHz, DMSO-*d*_6_): δ 170.9, 165.4, 150.2, 142.8, 142.7, 134.4, 132.2, 131.3, 126.0, 123.3, 122.6, 121.8, 119.6, 110.8, 101.8, 101.1, 67.2, 57.8, 45.9; HRMS (ESI): *m*/*z* calcd. for C_21_H_14_BrN_5_O_2_ [M + H]^+^ 448.0409, found 448.0425.

(±)-(3*R*,6′*R*)-8′-amino-1′,2-dioxo-6′-(4-nitrophenyl)-1′*H*,6′*H*-spiro[indoline-3,5′-pyrazolo[1,2-a]pyridazine]-7′-carbonitrile (**3q**) and (±)-(3*R*,6′*S*)-8′-amino-1′,2-dioxo-6′-(4-nitrophenyl)-1′*H*,6′*H*-spiro[indoline-3,5′-pyrazolo[1,2-a]pyridazine]-7′-carbonitrile (**3′q**)



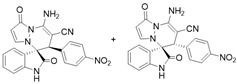



[Reaction time: 15.0 h], 108 mg, 26%, a yellow solid, m.p. 239.5–241.5 °C; IR (thin film): ν_max_ 3442, 3351, 3262, 2193, 1726, 1679, 1622, 1570, 1521, 1471, 1344, 1110, 857 cm^−1^; ^1^H NMR (400 MHz, DMSO-*d*_6_): δ **3q**: 10.57 (s, 1 H), 8.08 (d, *J* = 8.8 Hz, 2 H), 7.85 (s, 2 H), 7.75 (d, *J* = 7.6 Hz, 1 H), 7.61 (d, *J* = 4.0 Hz, 1 H), 7.40 (d, *J* = 4.0 Hz, 1 H), 7.28–7.23 (m, 1 H), 7.20 (ψt, *J* = 7.6 Hz, 1 H), 6.90 (ψt, *J* = 7.6 Hz, 1 H), 6.66 (d, *J* = 8.0 Hz, 1 H), 5.72 (d, *J* = 4.0 Hz, 1 H), 4.92 (s, 1 H); **3′q**: 10.97 (s, 1 H), 8.08 (d, *J* = 8.8 Hz, 2 H), 7.90 (s, 2 H), 7.34 (ψt, *J* = 7.6 Hz, 1 H), 7.28–7.23 (m, 4 H), 6.78 (d, *J* = 8.0 Hz, 1 H), 6.41 (d, *J* = 7.6 Hz, 1 H), 5.69 (d, *J* = 4.0 Hz, 1 H), 4.55 (s, 1 H); ^13^C NMR (100 MHz, DMSO-*d*_6_): δ 171.0, 170.7, 165.4 (2C), 150.4, 150.0, 147.6, 144.1, 143.7, 143.0, 142.9, 142.5, 142.0, 132.2, 131.6, 126.0, 125.6, 123.5, 123.4, 122.6, 122.4, 122.0, 119.5 (2C), 111.3, 110.9, 101.4, 101.3, 67.3, 67.0, 57.3, 57.2, 46.2, 45.7; HRMS (ESI): *m*/*z* calcd. forC_21_H_14_N_6_O_4_ [M + H]^+^ 415.1155, found 415.12316. 

(±)-(3*R*,6′*R*)-8′-amino-1′,2-dioxo-6′-(naphthalen-1-yl)-1′*H*,6′*H*-spiro[indoline-3,5′-pyrazolo[1,2-a]pyridazine]-7′-carbonitrile (**3r**)



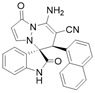



[Reaction time: 4.0 h], 140 mg, 28%, a yellow solid, m.p. 248.8–250.8 °C; IR (thin film): ν_max_ 3366, 3268, 3125, 2188, 1723, 1676, 1647, 1625, 1564, 1531, 1517, 1463, 1333, 1193, 743 cm^−1^; ^1^H NMR (400 MHz, DMSO-*d*_6_): δ 10.43 (s, 1 H), 7.84 (s, 4 H), 7.71 (s, 2 H), 7.54–7.20 (m, 7 H), 6.55 (d, *J* = 7.2 Hz, 1 H), 5.71 (s, 1 H), 4.89 (s, 1 H); ^13^C NMR (100 MHz, DMSO-*d*_6_): δ 171.2, 165.4, 150.4, 142.7 (2C), 132.9, 132.8, 132.5, 131.9, 129.6, 128.2, 127.9, 127.7 (2C), 126.8, 126.7, 126.1, 123.2, 122.9, 119.7, 110.7, 101.0, 67.4, 58.5, 46.7; HRMS (ESI): *m*/*z* calcd. for C_25_H_17_N_5_O_2_ [M + H]^+^ 420.1460, found 420.15148.

(±)-(3*R*,6′*R*)-8′-amino-1′,2-dioxo-6′-(3,4-dimethylphenyl)-1′*H*,6′*H*-spiro[indoline-3,5′-pyrazolo[1,2-a]pyridazine]-7′-carbonitrile (**3s**)



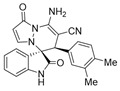



[Reaction time: 2.5 h], 216 mg, 54%, a white solid, m.p. 236.6–237.9 °C; IR (thin film): ν_max_ 3485, 3369, 3274, 3145, 2183, 1719, 1684, 1604, 1562, 1468, 1200, 752 cm^−1^; ^1^H NMR (400 MHz, DMSO-*d*_6_): δ 10.43 (s, 1 H), 7.73 (d, *J* = 7.6 Hz, 1 H), 7.69 (s, 2 H), 7.33–7.29 (m, 2 H), 7.17 (ψt, *J* = 7.6 Hz, 1 H), 6.88 (d, *J* = 6.8 Hz, 1 H), 6.75 (brs, 1 H), 6.64 (brs, 1 H), 6.63 (d, *J* = 8.0 Hz, 1 H), 5.66 (d, *J* = 4.0 Hz, 1 H), 4.58 (s, 1 H), 2.09 (s, 3 H), 2.03 (s, 3 H); ^13^C NMR (100 MHz, DMSO-*d*_6_): δ 171.2, 165.4, 150.4, 142.9, 142.5, 136.2, 135.8, 132.1, 131.8, 131.2, 129.3, 127.6, 126.0, 123.1, 123.0, 119.7, 110.7, 101.0, 67.4, 58.9, 46.1, 19.9, 19.4; HRMS (ESI): *m*/*z* calcd. for C_23_H_19_N_5_O_2_ [M + H]^+^ 398.1617, found 398.1625.

(±)-(3*R*,6′*S*)-8′-amino-6′-(3,4-dimethoxyphenyl)-1′,2-dioxo-1′H,6′H-spiro[indoline-3,5′-pyrazolo[1,2-a]pyridazine]-7′-carbonitrile (**3′t**)



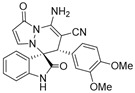



[Reaction time: 7.0 h], 171 mg, 43%, a white solid, m.p. 242.5–243.9 °C; IR (thin film): *ν*_max_ 3277, 3140, 2188, 1733, 1678, 1620, 1574, 1544, 1515, 1469, 1266, 1024, 755 cm^−1^; ^1^H NMR (400 MHz, DMSO-*d*_6_): δ 10.85 (s, 1 H), 7.79 (s, 2 H), 7.71 (d, *J* = 4.0 Hz, 1 H), 7.26 (d, *J* = 7.6 Hz, 1 H), 6.70 (ψt, *J* = 7.6 Hz, 1 H), 6.78–6.72 (m, 3 H), 6.56 (d, *J* = 7.6 Hz, 1 H), 6.20 (brs, 1 H), 5.64 (d, *J* = 4.0 Hz, 1 H), 4.30 (s, 1 H), 3.67 (s, 3 H), 3.34 (s, 3 H); ^13^C NMR (100 MHz, DMSO-*d*_6_): δ 171.0, 165.3, 149.8, 148.9, 148.0, 143.9, 141.8, 131.1, 126.9, 124.9, 123.6, 122.8, 122.4, 119.4, 113.3, 111.2 (2C), 100.9, 68.4, 59.0, 55.7, 55.5, 46.5; HRMS (ESI): *m*/*z* calcd. for C_23_H_20_N_5_O_4_ [M + H]^+^ 430.1515, found 430.15274.

(±)-(3*R*,6′*R*)-8′-amino-1′,2-dioxo-6′-(2,4-dichloroxyphenyl)-1′*H*,6′*H*-spiro[indoline-3,5′-pyrazolo[1,2-a]pyridazine]-7′-carbonitrile (**3u**) and (±)-(3*R*,6′*S*)-8′-amino-1′,2-dioxo-6′-(2,4-dichloroxyphenyl)-1′*H*,6′*H*-spiro[indoline-3,5′-pyrazolo[1,2-a]pyridazine]-7′-carbonitrile (**3′u**)



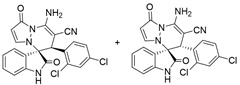



[Reaction time: 1.5 h], 382 mg, 87%, a yellow solid, m.p. 260.2–262.2 °C; IR (thin film): *ν*_max_ 3450, 3304, 2150, 2192, 1719, 1685, 1617, 1565, 1473, 1433, 1280, 1105, 837, 752 cm^−1^; ^1^H NMR (400 MHz, DMSO-*d*_6_): δ **3u**: 10.84 (s, 1 H), 7.86 (d, *J* = 7.2 Hz, 2 H), 7.51–7.46 (m, 2 H), 7.43–7.41 (m, 1 H), 7.39–7.36 (m, 2 H), 7.34–7.30 (m, 2 H), 6.77 (d, *J* = 8.0 Hz, 1 H), 5.67 (d, *J* = 4.0 Hz, 1 H), 4.93 (s, 1 H); **3′u**: 10.97 (s, 1 H), 7.86 (d, *J* = 7.2 Hz, 2 H), 7.56 (d, *J* = 7.2 Hz, 1 H), 7.51–7.46 (m, 2 H), 7.34–7.30 (m, 1 H), 7.11 (ψt, *J* = 7.6, 0.8 Hz, 1 H), 6.91 (d, *J* = 8.0 Hz, 1 H), 6.83 (ψt, *J* = 7.6, 0.8 Hz, 1 H), 5.86 (d, *J* = 7.2 Hz, 1 H), 5.68 (d, *J* = 4.0 Hz, 1 H), 4.38 (s, 1 H); ^13^C NMR (100 MHz, DMSO-*d*_6_): δ 171.8, 170.6, 165.4 (2C), 150.2, 150.1, 143.1, 142.9, 142.7, 142.0, 136.1, 135.4, 134.5, 124.0, 133.9, 132.7, 132.4, 132.1, 132.0, 131.9, 129.0, 128.8, 128.3, 128.0, 126.2, 125.7, 123.0, 122.9, 122.6, 120.9, 119.8, 119.3, 111.3, 110.9, 101.1, 100.8, 66.8, 65.8, 57.3, 56.2, 42.2, 41.5; HRMS (ESI): *m*/*z* calcd. for C_21_H_13_Cl_2_N_5_O_2_ [M + H]^+^ 438.0525, found 438.06128.

(±)-(3*R*,6′*R*)-8′-amino-1′,2-dioxo-6′-(3,4-dichlorophenyl)-1′*H*,6′*H*-spiro[indoline-3,5′-pyrazolo[1,2-a]pyridazine]-7′-carbonitrile (**3v**)



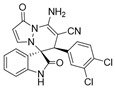



[Reaction time: 4.0 h], 183 mg, 42%, a yellow solid, m.p. 259.8–261.6 °C; IR (thin film): ν_max_ 3358, 3176, 3150, 2191, 1736, 1655, 1627, 1580, 1536, 1469, 1429, 1295, 1030, 830, 750 cm^−1^; ^1^H NMR (400 MHz, DMSO-*d*_6_): δ 10.61 (s, 1 H), 7.83 (s, 2 H), 7.71 (d, *J* = 6.4 Hz, 1 H), 7.48 (s, 1 H), 7.39–7.33 (m, 2 H), 7.19 (d, *J* = 6.4 Hz, 2 H), 6.94 (brs, 1 H), 6.70 (d, *J* = 6.8 Hz, 1 H), 5.71 (s, 1 H), 4.76 (s, 1 H); ^13^C NMR (100 MHz, DMSO-*d*_6_): δ 170.8, 165.4, 150.3, 143.0, 142.6, 136.2, 132.2, 132.0, 121.2, 130.9, 130.6, 126.0, 123.4, 122.4, 119.5, 110.9, 101.3, 67.0, 57.2, 45.5; HRMS (ESI): *m*/*z* calcd. for C_21_H_13_Cl_2_N_5_O_2_ [M + H]^+^ 438.0525, found 438.06011.

(±)-(3*R*,6′*R*)-8′-amino-1′,2-dioxo-6′-((E)-styryl)-1′*H*,6′*H*-spiro[indoline-3,5′-pyrazolo[1,2-a]pyridazine]-7′-carbonitrile (**3w**) and (±)-(3*R*,6′*S*)-8′-amino-1′,2-dioxo-6′-((E)-styry)-1′*H*,6′*H*-spiro[indoline-3,5′-pyrazolo[1,2-a]pyridazine]-7′-carbonitrile (**3′w**)



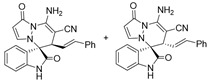



[Reaction time: 18 h], 246 mg, 62%, a yellow solid, m.p. 251.3–253.1 °C; IR (thin film): ν_max_ 3354, 3273, 3229, 2188, 1724, 1689, 1626, 1566, 1541, 1515, 1472, 1206, 1170, 11003, 968, 747 cm^−1^; ^1^H NMR (400 MHz, DMSO-*d*_6_): δ **3w**: 10.40 (s, 1 H), 7.74–7.56 (m, 3 H), 7.57 (d, *J* = 7.2 Hz, 1 H), 7.39–7.16 (m, 7 H), 6.96 (brs, 2 H), 6.60 (d, *J* = 7.6 Hz, 1 H), 5.67–5.65 (m, 1 H), 4.07–4.02 (m, 1 H); **3′w**: 10.78 (s, 1 H), 7.74–7.56 (m, 2 H), 7.39–7.16 (m, 9 H), 6.83 (d, *J* = 7.6 Hz, 1 H), 6.23 (d, *J* = 15.6 Hz, 1 H), 5.81–5.74 (m, 1 H), 5.67–5.65 (m, 1 H), 4.66 (s, 1 H); ^13^C NMR (100 MHz, DMSO-*d*_6_): δ 171.2, 171.1, 165.4, 165.3, 150.2, 149.3, 142.8, 142.6 (2C), 136.2, 135.8, 134.7, 131.8 (2C), 130.1 (2C), 129.2, 128.6, 128.4, 128.2, 126.7 (2C), 126.0, 125.6, 123.6, 123.5, 123.3, 123.1, 122.9, 119.8, 119.6, 110.9, 110.6, 101.0, 100.9, 100.5, 67.4, 66.2, 58.5, 57.4, 46.5, 44.4, 14.2; HRMS (ESI): *m*/*z* calcd. for C_23_H_17_N_5_O_2_ [M + H]^+^ 396.1460, found 396.1524.

(±)-(3*R*,6′*R*)-8′-amino-5-methyl-1′,2-dioxo-6′-phenyl-1′*H*,6′*H*-spiro[indoline-3,5′-pyrazolo[1,2-a]pyridazine]-7′-carbonitrile (**4a**)



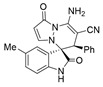



[Reaction time: 3.0 h], 307 mg, 80%, a white solid, m.p. 238.8–240.2 °C; IR (thin film): ν_max_ 3364, 3306, 3279, 3155, 2185, 1715, 1681, 1625, 1579, 1550, 1494, 1284, 1163, 742 cm^−1^; ^1^H NMR (400 MHz, DMSO-*d*_6_): δ 10.28 (s, 1 H), 7.69 (s, 2 H), 7.56 (s, 1 H), 7.31 (d, *J* = 4.0 Hz, 1 H), 7.15–7.09 (m, 4 H), 6.98 (brs, 2 H), 6.49 (d, *J* = 8.0 Hz, 1 H), 5.66 (d, *J* = 4.0 Hz, 1 H), 4.61 (s, 1 H), 2.34 (s, 3 H); ^13^C NMR (100 MHz, DMSO-*d*_6_): δ 171.1, 165.4, 150.1, 142.6, 140.4, 134.8, 132.1 (2C), 130.1, 128.4, 128.2, 126.4, 122.9, 119.6, 110.4, 100.9, 67.5, 58.5, 46.6, 21.3; HRMS (ESI): *m*/*z* calcd. for C_22_H_17_N_5_O_2_ [M+Na]^+^ 406.1280, found 406.13334.

(±)-(3*R*,6′*R*)-8′-amino-5-methoxy-1′,2-dioxo-6′-phenyl-1′*H*,6′*H*-spiro[indoline-3,5′-pyrazolo [1,2-a]pyridazine]-7′-carbonitrile (**4b**) and (±)-(3*R*,6′*S*)-8′-amino-5-methoxy-1′,2-dioxo-6′-phenyl-1′*H*,6′*H*-spiro[indoline-3,5′-pyrazolo[1,2-a]pyridazine]-7′-carbonitrile (**4′b**)



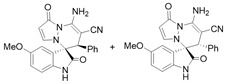



[Reaction time: 7.0 h], 280 mg, 70%, a brown solid, m.p. 239.6–241.2 °C; IR (thin film): ν_max_ 3363, 3245, 3214, 2172, 1732, 1683, 1624, 1576, 1539, 1494, 1431,1300, 1200, 781 cm^−1^; ^1^H NMR (400 MHz, DMSO-*d*_6_): δ **4b**: 10.21 (s, 1 H), 7.69 (s, 2 H), 7.44 (d, *J* = 2.4 Hz, 1 H), 7.36 (d, *J* = 4.0 Hz, 1 H), 7.28–7.20 (m, 2 H), 7.16–7.15 (m, 1H), 7.01 (brs, 2 H), 6.87–6.81 (m, 1 H), 6.52 (d, *J* = 8.4 Hz, 1 H), 5.67 (d, *J* = 4.0 Hz, 1 H), 4.70 (s, 1 H), 3.79 (s, 3 H); **4′b**: 10.97 (s, 1 H), 7.77 (s, 2 H), 7.55 (d, *J* = 4.0 Hz, 1 H), 7.28–7.20 (m, 2 H), 7.16–7.15 (m, 1 H), 6.96 (d, *J* = 6.8 Hz, 2 H), 6.87–6.81 (m, 1 H), 6.69 (d, *J* = 8.8 Hz, 1 H), 5.83 (d, *J* = 1.6 Hz, 1 H), 5.64 (d, *J* = 4.0 Hz, 1 H), 4.22 (s, 1 H), 3.49 (s, 3 H); ^13^C NMR (100 MHz, DMSO-*d*_6_): δ 171.2, 171.1, 165.4, 165.3, 155.9, 154.7, 150.1, 149.7, 143.4, 142.7, 136.5, 135.9, 135.2, 134.8, 130.1, 128.6, 128.4 (2C), 128.3, 123.9, 123.5, 119.8, 119.6, 117.1, 115.9, 112.4, 112.3, 111.6, 111.3, 100.9, 100.8, 67.8, 58.5, 56.5, 56.1, 55.7, 46.4; HRMS (ESI): *m*/*z* calcd. for C_22_H_17_N_5_O_3_ [M+K]^+^ 438.0968, found 438.10269.

(±)-(3*R*,6′*R*)-8′-amino-5-fluoro-1′,2-dioxo-6′-phenyl-1′*H*,6′*H*-spiro[indoline-3,5′-pyrazolo[1,2-a]pyridazine]-7′-carbonitrile (**4c**)



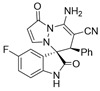



[Reaction time: 8.5 h], 237 mg, 61%, a white solid, m.p. 231.4–233.4 °C; IR (thin film): ν_max_ 3431, 3270, 3146, 2189, 1727, 1658, 1622, 1574, 1543, 1488, 1428, 1275, 1175, 1071, 802 cm^−1^; ^1^H NMR (400 MHz, DMSO-*d*_6_): δ 10.43 (s, 1 H), 7.75–7.70 (m, 2 H), 7.48 (d, *J* = 4.0 Hz, 1 H), 7.18–7.12 (m, 4 H), 7.00 (brs, 2 H), 6.60 (dd, *J* = 8.8, 4.4 Hz, 1 H), 5.71 (d, *J* = 4.0 Hz, 1 H), 4.69 (s, 1 H); ^13^C NMR (100 MHz, DMSO-*d*_6_): δ 171.1, 165.4, 150.1, 143.0, 139.1 (d), 134.5, 130.1, 128.5, 128.4, 124.4 (2C), 119.5, 118.4 (d), 114.2, 114.0, 111.7 (d), 101.2, 67.6, 58.3, 46.5; ^19^F NMR (376 MHz, DMSO-*d*_6_): δ -119.6; ^19^F NMR (376 MHz, DMSO-*d*_6_): δ −119.6; HRMS (ESI): *m*/*z* calcd. for C_21_H_14_FN_5_O_2_ [M+Na]^+^ 410.1029, found 410.10951.

(±)-(3*R*,6′*R*)-8′-amino-5-chloro-1′,2-dioxo-6′-phenyl-1′*H*,6′*H*-spiro[indoline-3,5′-pyrazolo[1,2-a]pyridazine]-7′-carbonitrile (**4d**)



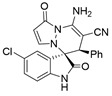



[Reaction time: 6.0 h], 331 mg, 82%, a brown solid, m.p. 231.0–232.9 °C; IR (thin film): ν_max_ 3435, 3270, 3141, 2191, 1722, 1654, 1627, 1559, 1527, 1477, 1427, 1276, 1211, 909, 703 cm^−1^; ^1^H NMR (400 MHz, DMSO-*d*_6_): δ 10.54 (s, 1 H), 7.90 (d, *J* = 1.6 Hz, 1 H), 7.70 (d, *J* = 5.6 Hz, 1 H), 7.51 (d, *J* = 4.0 Hz, 1 H), 7.36–7.33 (m, 1 H), 7.18–7.15 (m, 3 H), 6.98 (brs, 2 H), 6.61 (d, *J* = 7.6 Hz, 1 H), 5.57 (d, *J* = 4.0 Hz, 1 H), 4.71 (s, 1 H); ^13^C NMR (100 MHz, DMSO-*d*_6_): δ 170.9, 165.4, 150.1, 143.0, 141.8, 134.5, 131.8, 130.1, 128.6, 128.4, 127.1, 126.4, 124.8, 119.5, 112.1, 101.2, 67.4, 58.3, 46.4; HRMS (ESI): *m*/*z* calcd. for C_21_H_14_ClN_5_O_2_ [M + H]^+^ 404.0914, found 404.09140.

(±)-(3*R*,6′*R*)-8′-amino-6-bromo-1′,2-dioxo-6′-phenyl-1′*H*,6′*H*-spiro[indoline-3,5′-pyrazolo[1,2-a]pyridazine]-7′-carbonitrile (**4e**)



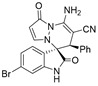



[Reaction time: 12 h], 213 mg, 47%, a brown solid, m.p. 230.8–232.7 °C; IR (thin film): ν_max_ 3426, 3395, 3365, 3138, 2188, 1733, 1682, 1655, 1618, 1577, 1543, 1482, 1455, 1278, 1175, 742, 593 cm^−1^; ^1^H NMR (400 MHz, DMSO-*d*_6_): δ 10.57 (s, 1 H), 7.71 (s, 2 H), 7.69 (s, 1 H), 7.48 (d, *J* = 4.0 Hz, 1 H), 7.37 (d, *J* = 8.0 Hz, 1 H), 7.19 (s, 3 H), 6.98 (brs, 2 H), 6.77 (s, 1 H), 5.69 (d, *J* = 4.0 Hz, 1 H), 4.68 (s, 1 H); ^13^C NMR (100 MHz, DMSO-*d*_6_): δ 170.9, 165.4, 150.1, 143.0, 141.8, 134.5, 131.8, 130.1, 128.6, 128.4, 127.1, 126.4, 124.8, 119.5, 112.1, 101.2, 67.4, 58.3, 46.4; HRMS (ESI): *m*/*z* calcd. for C_21_H_14_BrN_5_O_2_ [M + H]^+^ 448.0409, found 448.04897.

(±)-(3*R*,6′*R*)-8′-amino-5-iodo-1′,2-dioxo-6′-phenyl-1′*H*,6′*H*-spiro[indoline-3,5′-pyrazolo[1,2 -a]pyridazine]-7′-carbonitrile (**4f**)



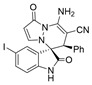



[Reaction time: 8.0 h], 207 mg, 42%, a brown solid, m.p. 234.2–235.6 °C; IR (thin film): ν_max_ 3144, 2190, 1735, 1687, 1616, 1568, 1544, 1468, 1427, 1319, 1206, 750, 527 cm^−1^; ^1^H NMR (400 MHz, DMSO-*d*_6_): δ 10.51 (s, 1 H), 8.12 (d, *J* = 1.6 Hz, 1 H), 7.70 (s, 2 H), 7.63 (dd, *J* = 8.0, 1.6 Hz, 1 H), 7.47 (d, *J* = 4.0 Hz, 1 H), 7.18–7.16 (m, 3 H), 6.98 (brs, 2 H), 6.44 (d, *J* = 8.0 Hz, 1 H), 5.69 (d, *J* = 4.0 Hz, 1 H), 4.69 (s, 1 H); ^13^C NMR (100 MHz, DMSO-*d*_6_): δ 170.6, 165.3, 150.1, 143.0, 142.6, 140.3, 134.5, 134.4, 130.0, 128.5, 128.4, 125.3, 119.5, 112.9, 101.1, 85.8, 67.2, 58.3, 46.3; HRMS (ESI): *m*/*z* calcd. for C_21_H_14_IN_5_O_2_ [M + H]^+^ 496.0270, found 496.03396.

(±)-(3*R*,6′*R*)-8′-amino-5-nitro-1′,2-dioxo-6′-phenyl-1′*H*,6′*H*-spiro[indoline-3,5′-pyrazolo[1,2-a]pyridazine]-7′-carbonitrile (**4g**)



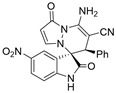



[Reaction time: 4.0 h], 83 mg, 20%, a yellow solid, m.p. 234.5–236.5 °C; IR (thin film): ν_max_ 3466, 3376, 3344, 3134, 2193, 1745, 1683, 1622, 1560, 1526, 1481, 1427, 1279, 1176, 841 cm^−1^; ^1^H NMR (400 MHz, DMSO-*d*_6_): δ 11.10 (s, 1 H), 8.73 (d, *J* = 6.0 Hz, 1 H), 8.25 (dd, *J* = 8.8, 2.0 Hz, 1 H), 7.74 (s, 2 H), 6.98 (brs, 2 H), 6.79 (d, *J* = 8.8 Hz, 2 H), 5.76 (d, *J* = 4.0 Hz, 1 H), 4.87 (s, 1 H); ^13^C NMR (100 MHz, DMSO-*d*_6_): δ 171.5, 165.4, 150.1, 149.1, 143.4, 143.2, 134.1, 130.1, 128.8, 128.7, 128.5, 123.9, 122.6, 119.4, 111.0, 101.4, 67.1, 58.0, 46.3; HRMS (ESI): *m*/*z* calcd. for C_21_H_14_N_6_O_4_ [M + H]^+^ 415.1155, found 415.12022.

(±)-(3*R*,6′*S*)-8′-amino-1′,2-dioxo-6′-phenyl-7-(trifluoromethyl)-1′*H*,6′*H*-spiro[indoline-3,5′-pyrazolo[1,2-a]pyridazine]-7′-carbonitrile (**4′h**) and (±)-(3*R*,6′*R*)-8′-amino-1′,2-dioxo-6′-phenyl-7-(trifluoromethyl)-1′*H*,6′*H*-spiro[indoline-3,5′-pyrazolo[1,2-a]pyridazine]-7′-carbonitrile (**4h**)



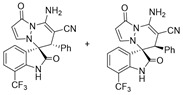



[Reaction time: 6.0 h], 324 mg, 74%, a yellow solid, m.p. 235.4–236.7 °C; IR (thin film): ν_max_ 3456, 3280, 3121, 2187, 1736, ^1^H NMR (400 MHz, DMSO-*d*_6_): δ **4′h**: 11.37 (s, 1 H), 7.83 (s, 2 H), 7.67 (d, *J* = 4.0 Hz, 1 H), 7.55 (d, *J* = 8.4 Hz, 1 H), 7.28–7.14 (m, 3 H), 7.05 (ψt, *J* = 8.0 Hz, 1 H), 6.92 (d, *J* = 6.4 Hz, 2 H), 6.56 (d, *J* = 7.6 Hz, 1 H), 5.74 (d, *J* = 4.0 Hz, 1 H), 4.35 (s, 1 H); **4h**: 10.91 (s, 1 H), 8.03 (d, *J* = 7.6 Hz, 1 H), 7.77 (s, 2 H), 7.60 (d, *J* = 8.0 Hz, 1 H), 7.55 (d, *J* = 8.4 Hz, 1 H), 7.36 (ψt, *J* = 7.6 Hz, 1 H), 7.28–7.14 (m, 3 H), 6.92 (d, *J* = 6.4 Hz, 2 H), 5.70 (d, *J* = 4.0 Hz, 1 H), 4.71 (s, 1 H); ^13^C NMR (100 MHz, DMSO-*d*_6_): δ 172.0, 171.6, 165.4, 165.3, 150.2, 149.7, 150.1, 143.7, 143.2, 139.5, 135.7, 134.0, 129.9, 129.4, 128.7, 128.6, 128.4, 128.3, 127.7, 124.9, 124.8, 124.5, 123.3, 122.5, 122.1, 119.6, 112.2, 111.9, 101.3, 101.1, 66.4, 66.3, 58.1, 58.0, 46.9, 46.3; ^19^F NMR (376 MHz, DMSO-*d*_6_): δ −60.2, −60.4; HRMS (ESI): *m*/*z* calcd. for C_22_H_14_F_3_N_5_O_2_ [M + H]^+^ 438.1178, found 438.1179.

(±)-(3*R*,6′*R*)-8′-amino-2′-methyl-1′,2-dioxo-6′-phenyl-1′*H*,6′*H*-spiro[indoline-3,5′-pyrazolo[1,2-a]pyridazine]-7′-carbonitrile (**5a**)



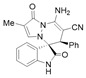



[Reaction time: 1.0 h], 299 mg, 78%, a yellow solid, m.p. 257.5–259.4 °C; IR (thin film): ν_max_ 3465, 3276, 3095, 2189, 1733, 1679, 1614, 1575, 1468, 1425, 1245, 1160, 746 cm^−1^; ^1^H NMR (400 MHz, DMSO-*d*_6_): δ 10.34 (s, 1 H), 7.73 (d, *J* = 7.6 Hz, 1 H), 7.67 (s, 2 H), 7.30 (ψt, *J* = 7.6 Hz, 1 H), 7.18–7.12 (m, 4 H), 6.97 (brs, 2 H), 6.60 (d, *J* = 7.6 Hz, 1 H), 4.63 (s, 1 H), 1.69 (s, 3 H); ^13^C NMR (100 MHz, DMSO-*d*_6_): δ 171.2, 165.7, 150.8, 142.8, 139.6, 135.0, 131.7, 130.1, 128.3, 128.2, 125.9, 123.3, 123.0, 119.7, 110.2, 100.6, 67.3, 58.7, 46.7, 7.5; HRMS (ESI): *m*/*z* calcd. for C_22_H_17_N_5_O_2_ [M+Na]^+^ 406.1280, found 406.13367.

(±)-(3*R*,6′*S*)-8′-amino-2′-methyl-1′,2-dioxo-6′-(*p*-tolyl)-1′*H*,6′*H*-spiro[indoline-3,5′-pyrazolo[1,2-a]pyridazine]-7′-carbonitrile (**5′b**) and (±)-(3*R*,6′*R*)-8′-amino-2′-methyl-1′,2-dioxo-6′-(*p*-tolyl)-1′*H*,6′*H*-spiro[indoline-3,5′-pyrazolo[1,2-a]pyridazine]-7′-carbonitrile (**5b**)



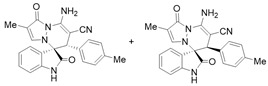



[Reaction time: 3.5 h], 346 mg, 87%, a white solid, m.p. 240.1–242.0 °C; IR (thin film): ν_max_ 3376, 3210, 3101, 2193, 1717, 1673, 1624, 1570, 1515, 1473, 1427, 1250, 1161, 1047, 739 cm^−1^; ^1^H NMR (400 MHz, DMSO-*d*_6_): δ **5′b**: 10.84 (s, 1 H), 7.76 (s, 2 H), 7.43 (s, 1 H), 7.22 (ψt, *J* = 7.6 Hz, 1 H), 6.96–6.90 (m, 2 H), 6.85 (brs, 1 H), 6.78 (d, *J* = 7.2 Hz, 2 H), 6.71 (d, *J* = 7.6 Hz, 1 H), 6.57 (d, *J* = 7.6 Hz, 1 H), 4.27 (s, 1 H), 2.20 (s, 3 H), 1.65 (s, 3 H); **5b**: 10.35 (s, 1 H), 7.72 (d, *J* = 7.6 Hz, 1 H), 7.65 (s, 2 H), 7.30 (ψt, *J* = 7.6 Hz, 1 H), 7.16 (ψt, *J* = 7.6 Hz, 1 H), 7.11 (s, 1 H), 6.96–6.90 (m, 4 H), 6.62 (d, *J* = 8.0 Hz, 1 H), 4.59 (s, 1 H), 2.18 (s, 3 H), 1.69 (s, 3 H); ^13^C NMR (100 MHz, DMSO-*d*_6_): δ 171.4, 171.2, 165.6, 150.0, 142.8, 141.8, 140.5, 139.5, 137.6, 137.4, 132.6, 131.9, 131.6, 131.0, 130.0, 128.8, 125.9, 125.2, 123.4, 123.0, 122.3, 119.7, 119.6, 111.2, 111.0 (2C), 110.6, 68.1, 67.3, 59.0, 58.9, 46.3, 46.2, 21.1 (2C), 19.0, 7.5; HRMS (ESI): *m*/*z* calcd. for C_23_H_19_N_5_O_2_ [M + H]^+^ 398.1617, found.398.16345.

(±)-(3*R*,6′*R*)-8′-amino-6′-(4-methoxylphenyl)-2′-methyl-1′,2-dioxo-1′*H*,6′*H*-spiro[indoline-3,5′-pyrazolo[1,2-a]pyridazine]-7′-carbonitrile (**5c**) and (±)-(3*R*,6′*S*)-8′-amino-6′-(4-methoxylphenyl)-2′-methyl-1′,2-dioxo-1′*H*,6′*H*-spiro[indoline-3,5′-pyrazolo[1,2-a]pyridazinee]-7′-carbonitrile (**5′c**)



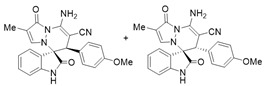



[Reaction time: 4.0 h], 290 mg, 73%, a yellow solid, m.p. 241.5–242.7 °C; IR (thin film): ν_max_ 3369, 3259, 3208, 2178, 1716, 1672, 1624, 1598, 1567, 1510, 1470, 1428, 1247, 1175, 738 cm^−1^; ^1^H NMR (400 MHz, DMSO-*d*_6_): δ **5c**: 10.35 (s, 1 H), 7.75 (s, 1 H), 7.65 (s, 2 H), 7.31 (ψt, *J* = 7.6 Hz, 1 H), 7.16 (ψt, *J* = 7.6 Hz, 1 H), 7.11 (s, 1 H), 6.94–6.80 (m, 3 H), 6.74–6.63 (m, 1 H), 6.63 (d, *J* = 7.6 Hz, 1 H), 4.58 (s, 1 H), 3.65 (s, 3 H), 1.69 (s, 3 H); **5′c**: 10.84 (s, 1 H), 7.71 (d, *J* = 7.2 Hz, 2 H), 7.43 (s, 1 H), 7.23 (ψt, *J* = 7.6 Hz, 1 H), 6.94–6.80 (m, 3 H), 6.74–6.63 (m, 3 H), 6.56 (d, *J* = 7.2 Hz, 1 H), 4.25 (s, 1 H), 3.67 (s, 3 H), 1.65 (s, 3 H); ^13^C NMR (100 MHz, DMSO-*d*_6_): δ 171.3, 165.6, 159.1, 150.0 (2C), 142.8, 141.8, 140.5, 139.5, 131.6, 131.2, 131.0, 127.4, 126.6, 125.9, 125.2, 123.4, 123.0, 122.4, 119.8, 119.6, 113.6, 111.1, 111.0, 110.9, 110.6, 68.2, 67.3, 59.1 (2C), 55.4 (2C), 45.9, 45.8, 19.0, 7.5; HRMS (ESI): *m*/*z* calcd. for C_23_H_19_N_5_O_3_ [M + H]^+^ 414.1566, found 414.12751.

(±)-(3*R*,6′*R*)-8′-amino-6′-(4-fluorophenyl)-2′-methyl-1′,2-dioxo-1′*H*,6′*H*-spiro[indoline-3,5′-pyrazolo[1,2-a]pyridazine]-7′-carbonitrile (**5d**)



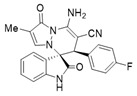



[Reaction time: 2.0 h], 297 mg, 91%, a white solid, m.p. 260.8–262.4 °C; IR (thin film): ν_max_ 3372, 3333, 3213, 3104, 2181, 1720, 1670, 1628, 1590, 1509, 1476, 1427, 1251, 1191, 737 cm^−1^; ^1^H NMR (400 MHz, DMSO-*d*_6_): δ 10.39 (s, 1 H), 7.71 (d, *J* = 7.2 Hz, 1 H), 7.69 (s, 2 H), 7.32 (ψt, *J* = 8.0, 1.2 Hz, 1 H), 7.17 (ψt, *J* = 8.8, 0.8 Hz, 1 H), 7.14 (d, *J* = 1.2 Hz, 1 H), 6.99 (d, *J* = 6.0 Hz, 4 H), 6.64 (d, *J* = 7.6 Hz, 1 H), 4.67 (s, 1 H), 1.69 (d, *J* = 0.8 Hz, 3 H); ^13^C NMR (100 MHz, DMSO-*d*_6_): δ 171.1, 165.7, 150.1, 142.7, 139.6, 132.1, 132.0, 131.8, 131.2, 125.9, 123.2 (2C), 119.7, 115.2, 115.0, 111.2, 110.6, 67.2, 58.4, 45.8, 7.5; ^19^F NMR (376 MHz, DMSO-*d*_6_): −114.3; HRMS (ESI): *m*/*z* calcd. for C_22_H_16_FN_5_O_2_ [M+K]^+^ 440.0925, found 440.09885.

(±)-(3*R*,6′*R*)-8′-amino-6′-(4-chlorophenyl)-2′-methyl-1′,2-dioxo-1′*H*,6′*H*-spiro[indoline-3,5′-pyrazolo[1,2-a]pyridazine]-7′-carbonitrile (**5e**)



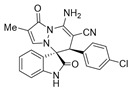



[Reaction time: 3.5 h], 243 mg, 58%, a yellow solid, m.p. 260.1–262.0 °C; IR (thin film): ν_max_ 3472, 3373, 3344, 2182, 1725, 1652, 1624, 1567, 1471, 1427, 1277, 1161, 832, 742 cm^−1^; ^1^H NMR (400 MHz, DMSO-*d*_6_): δ 10.41 (s, 1 H), 7.72 (s, 1 H), 7.71 (s, 2 H), 7.32 (ψt, *J* = 7.6 Hz, 1 H), 7.24–7.14 (m, 4 H), 6.99 (brs, 2 H), 6.66 (d, *J* = 7.6 Hz, 1 H), 4.68 (s, 1 H), 1.69 (s, 3 H); ^13^C NMR (100 MHz, DMSO-*d*_6_): δ 171.0, 165.7, 150.1, 142.7, 139.7, 134.2, 133.1, 131.9, 131.8, 128.3, 125.9, 123.2, 123.0, 119.7, 111.3, 110.7, 67.1, 58.1, 45.8, 7.5; HRMS (ESI): *m*/*z* calcd. for C_22_H_16_ClN_5_O_2_ [M + H]^+^ 418.1071, found 429.11331.

(±)-(3*R*,6′*R*)-8′-amino-6′-(4-bromophenyl)-2′-methyl-1′,2-dioxo-1′*H*,6′*H*-spiro[indoline-3,5′ -pyrazolo[1,2-a]pyridazine]-7′-carbonitrile (**5f**)



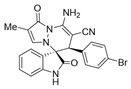



[Reaction time: 1.5 h], 305 mg, 66%, a yellow solid, m.p. 263.7–265.7 °C; IR (thin film): ν_max_ 3472, 3342, 3190, 3131, 3098, 2188, 1733, 1651, 1619, 1561, 1472, 1413, 1277, 1157, 744, 516 cm^−1^; ^1^H NMR (400 MHz, DMSO-*d*_6_): δ 10.41 (s, 1 H), 7.72 (s, 1 H), 7.70 (s, 1 H), 7.32 (ψt, *J* = 7.6 Hz, 1 H), 7.23 (d, *J* = 7.6 Hz, 1 H), 7.17 (ψt, *J* = 7.6 Hz, 1 H), 7.15 (s, 1 H), 6.97 (s, 2 H), 6.65 (d, *J* = 8.0 Hz, 1 H), 4.68 (s, 1 H), 1.69 (s, 3 H); ^13^C NMR (100 MHz, DMSO-*d*_6_): δ 171.1, 165.7, 150.1, 142.7, 139.7, 134.6, 132.2, 131.8, 131.3, 125.9, 123.2, 123.0, 121.8, 119.7, 111.3, 110.7, 67.0, 58.0, 45.9, 7.5; HRMS (ESI): *m*/*z* calcd. for C_22_H_16_BrN_5_O_2_ [M + H]^+^ 462.0566, found 462.06279.

(±)-(3*R*,6′*S*)-8′-amino-6′-(4-nitrophenyl)-2′-methyl-1′,2-dioxo-1′*H*,6′*H*-spiro[indoline-3,5′-pyrazolo[1,2-a]pyridazine]-7′-carbonitrile (**5′g**) and (±)-(3*R*,6′*R*)-8′-amino-6′-(4-nitrophen-yl)-2′-methyl-1′,2-dioxo-1′*H*,6′*H*-spiro[indoline-3,5′-pyrazolo[1,2-a]pyridazine]-7′-carbonitrile (**5g**)



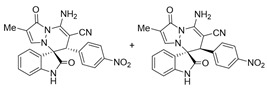



[Reaction time: 16.0 h], 301 mg, 70%, a yellow solid, m.p. 261.7–263.3 °C; IR (thin film): ν_max_ 3368, 3260, 3217, 2188, 1722, 1675, 1619, 1571, 1521, 1473, 1428, 1251, 1165, 848, 748 cm^−1^; ^1^H NMR (400 MHz, DMSO-*d*_6_): δ **5′g**: 10.94 (s, 1 H), 8.09 (s, 2 H), 7.87 (s, 2 H), 7.38–7.19 (m, 4 H), 6.88 (ψt, *J* = 7.6 Hz, 1 H), 6.77 (d, *J* = 8.0 Hz, 1 H), 6.66 (d, *J* = 7.6 Hz, 1 H), 4.51 (s, 1 H), 1.66 (s, 3 H); **5g**: 10.50 (s, 1 H), 8.07 (s, 2 H), 7.81 (s, 2 H), 7.74 (d, *J* = 7.6 Hz, 1 H), 7.38–7.19 (m, 5 H), 6.36 (d, *J* = 7.2 Hz, 1 H), 4.89 (s, 1 H), 1.70 (s, 3 H); ^13^C NMR (100 MHz, DMSO-*d*_6_): δ 171.3, 170.7, 165.7, 165.7, 150.3, 149.9, 147.6, 144.4, 143.2, 142.5, 142.0, 140.3, 139.8, 132.0, 131.5, 131.4, 125.9, 125.6, 123.4, 123.3, 122.8, 122.5, 122.0, 119.6, 111.6, 111.3, 111.2, 110.8, 67.2, 67.0, 57.5, 57.4, 46.2, 45.8, 21.2, 7.5; HRMS (ESI): *m*/*z* calcd. for C_22_H_16_N_6_O_4_ [M + H]^+^ 429.1311, found 429.13739.

(±)-Ethyl 2-(((3R,6′R)-8′-amino-7′-cyano-1′,2-dioxo-6′-phenyl-1′H,6′H-spiro[indoline-3,5′-pyrazolo[1,2-a]pyridazin]-1-yl)methyl)acrylate (**6**)



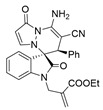



[Reaction time: 5.0 min], 73 mg, 76%, a white solid m.p. 206.0–207.3 °C; IR (thin film): ν_max_ 3370, 2981, 2189, 1725, 1683, 1492, 1382, 1172, 1027, 754 cm^−1^; ^1^H NMR (400 MHz, DMSO-*d*_6_): δ 7.85 (d, *J* = 7.2 Hz, 1 H), 7.72 (s, 2 H), 7.40 (td, *J* = 8.0–7.6, 0.8 Hz, 1 H), 7.32 (d, *J* = 4.0 Hz, 1 H), 7.28 (ψt, *J* = 7.6–7.2 Hz 1 H), 7.20–7.30 (m, 3 H), 6.91 (brs, 2 H), 6.77 (d, *J* = 7.8 Hz, 1 H), 5.76 (s, 1 H), 5.67 (d, *J* = 4.0 Hz, 1 H), 4.77 (s, 1 H), 4.20 (d, *J* = 16.8 Hz, B of AB, 1 H), 4.14–4.04 (m, 2 H), 1.18 (t, *J* = 7.2 Hz, 3 H); ^13^C NMR (100 MHz, DMSO-*d*_6_): δ 169.3, 165.4, 165.0, 150.2, 143.3, 142.8, 134.4, 133.3, 132.0, 130.2, 128.6, 128.4, 126.0, 125.2, 124.1, 122.0, 119.5, 110.6, 101.4, 67.1, 61.2, 58.5, 46.4, 14.4; HRMS (ESI): *m*/*z* calcd. for C_27_H_24_N_5_O_4_ [M + H]^+^ 482.1828, found 482.1689.

(±)-*tert*-butyl (3*R*,6′*R*)-8′-(bis(tert-butoxycarbonyl)amino)-7′-cyano-1′,2-dioxo-6′-phenyl-1′H,6′H-spiro[indoline-3,5′-pyrazolo[1,2-a]pyridazine]-1-carboxylate (**7**)



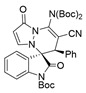



[Reaction time: 5.0 h], 95 mg, 71%, a white solid, m.p. 159.2–160.3 °C; IR (thin film): ν_max_ 2980, 2926, 2855, 2221, 1775, 1741, 1708, 1647, 1551, 1465, 1043, 914, 857 cm^−1^; ^1^H NMR (400 MHz, DMSO-*d*_6_): δ 7.81 (d, *J* = 7.6 Hz, 1 H), 7.57 (ψt, *J* = 7.6 Hz, 1 H), 7.53 (ψt, *J* = 7.6 Hz, 1 H), 7.47–7.43 (m, 2 H), 7.28–7.19 (m, 3 H), 6.83 (s, 2 H), 5.75 (d, *J* = 4.0 Hz, 1 H), 4.91 (s, 1 H), 1.60 (s, 9 H), 1.44 (s, 9 H), 1.38 (s, 9 H); ^13^C NMR (100 MHz, DMSO-*d*_6_): δ 1670.1, 162.3, 148.4, 147.8 (2C), 145.3, 140.4, 139.5, 132.5, 132.0, 129.8, 129.2, 128.6, 126.2, 125.9, 121.1, 115.4, 115.1, 103.7, 91.9, 84.5 (2C), 84.2, 67.2, 66.6, 48.9, 29.8, 28.2, 27.9, 27.8; HRMS (ESI): *m*/*z* calcd. for C_36_H_39_N_5_NaO_8_ [M+Na]^+^ 692.2696, found 692.2679.

(±)-(3R,6′S)-1′,2,8′-trioxo-6′-(o-tolyl)-7′,8′-dihydro-1′H,6′H-spiro[indoline-3,5′-pyrazolo[1,2-a]pyridazine]-7′-carbonitrile (**8**)



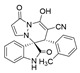



[Reaction time: 5.0 h], 19 mg, 25%, a yellow solid, m.p. 270.7–272.7 °C; IR (thin film): ν_max_ 3533, 3365, 3175, 2923, 2852, 2187, 1751, 1718, 1677, 1577, 1430, 1343, 1169, 1014, 887, 740 cm^−1^; ^1^H NMR (400 MHz, DMSO-*d*_6_): δ 10.90 (s, 1 H), 8.37 (s, 1 H), 7.34–7.30 (m, 2 H), 7.28–7.19 (m, 2 H), 7.08 (d, *J* = 6.4 Hz, 1 H), 7.01 (d, *J* = 6.8 Hz, 1 H), 6.90 (d, *J* = 7.6 Hz, 1 H), 6.72 (ψt, *J* = 7.2 Hz, 1 H), 5.74 (d, *J* = 7.2 Hz, 1 H), 5.64 (d, *J* = 4.0 Hz, 1 H), 4.44 (s, 1 H), 1.71 (s, 3 H), 1.70 (s, 3 H); ^13^C NMR (100 MHz, DMSO-*d*_6_): δ 172.3, 161.8, 143.8, 143.0, 138.8, 133.5, 131.8, 130.7, 130.1, 129.0, 128.4, 126.6, 126.2, 122.3, 121.2, 118.4, 111.0, 101.1, 89.2, 65.7, 41.9, 19.3; HRMS (ESI): *m*/*z* calcd. for C_22_H_16_N_4_NaO_3_ [M+Na]^+^ 407.1120, found 407.0876.

(±)-(3R,6′R)-1-acetyl-8′-amino-2′-methyl-1′,2-dioxo-6′-phenyl-1′H,6′H-spiro[indoline-3,5′-pyrazolo[1,2-a]pyridazine]-7′-carbonitrile (**9**)



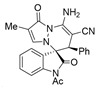



[Reaction time: 5.0 h], 73 mg, 86%, a white solid, m.p. 252.5–254.4 °C; IR (thin film): ν_max_ 3414, 3015, 2924, 2219, 1728, 1666, 1626, 1571, 1529, 1471, 1372, 1197, 897, 755 cm^−1^; ^1^H NMR (400 MHz, DMSO-*d*_6_): δ 7.88–7.78 (m, 4 H), 7.51–7.48 (m, 2 H), 7.25 (s, 1 H), 7.20–7.16 (m, 3 H), 6.84 (brs, 2 H), 4.67 (s, 1 H), 2.29 (s, 3 H), 1.70 (s, 3 H); ^13^C NMR (100 MHz, DMSO-*d*_6_): δ 170.6, 169.6, 165.6, 150.1, 140.4, 140.1, 133.7, 129.6, 131.4, 125.9, 125.6, 123.4, 123.3, 122.8, 122.5, 122.0, 119.6, 111.6, 111.3, 111.2, 110.8, 67.2, 67.0, 57.5, 57.4, 46.2, 45.8, 21.2, 7.5; HRMS (ESI): *m*/*z* calcd. for C_24_H_19_N_5_NaO_3_ [M+Na]^+^ 448.1386, found 448.1376.

## 4. Conclusions

In summary, we have developed an abnormally formal [3+3]-cycloaddition of *N*-unsubstituted isatin *N*,*N*′-cyclic azomethine imine 1,3-dipoles with Knoevenagel adducts to form novel dicyclic spiropyridazine oxindole derivatives in moderate to excellent yields (20–93%) and low to high diastereoselectivities (1:9–10:1 *dr*) under the optimized reaction condition. All the synthesized spiro-oxindole derivatives **3**/**3′**, **4**/**4′**, and **5**/**5′** were confirmed by ^1^H and ^13^C NMR, IR, and HMRS technologies. The relative stereochemistry of products was determined by the single-crystal X-ray of **4a** and the single-crystal X-ray of the compound reported by Moghaddam [14].

## Data Availability

Not applicable.

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
