# Peer review of "K2CO3-Promoted Formal [3+3]-Cycloaddition of N-Unsubstituted Isatin N,N′-Cyclic Azomethine Imine 1,3-Dipoles with Knoevenagel Adducts"

_molecules, 2023, doi:10.3390/molecules28031034_

Round 1
Reviewer 1 Report
This manuscript describes the synthesis of dicyclic spiropyridazine oxoindole derivatives, using [3+3]-cycloaddition of N-unsubstituted isatin N,N’-cyclic azomethine imine 1,3-dipoles, generated from the condensation of isatins and pyrazolidones, with Knoevenagel adducts. This seems like a well written and thorough study.
There’s one more question, why DABCO was so efficient for the cycloaddition with the N-substituted isatin N,N’-cyclic azomethine imine 1,3-dipoles, while was negative for the N-unsubstituted one. The author didn’t give an answer in the paper.
This manuscript is recommended to publish in Molecules.
Reviewer 2 Report
The authors describe formal [3+3]‐cycloaddition of N‐unsubstituted isatin N,N’‐cyclic azomethine imine 1,3‐dipoles and Knoevenagel adducts promoted with K2CO3. This work is very similar to the previously published article by Matloubi Mogaddam and co‐workers [ref. 13] about of N‐substituted isatin N,N’‐cyclic azomethine imine 1,3‐dipoles. In this regard, the novelty of this article is not very high. However, due to the large amount of work done, it may be published after the major revision. Some comments are listed below:
There are often extra spaces in abbreviations and compound names.
Line 33, Figure 1. camptothecin minics. Do you mean . camptothecin mimics?
Lines 44 ...check spelling (diploarophile, phenyldenemalononitrile). See attached file (highlighted in red).
Figure 1. ACC inhibitor should be III
Scheme 1. Do not specify the configuration for 6' or provide the structure of another diastereomer
LIne 59. imines become be urgent. Check grammar
LIne 77. reacted smoothly with 2a and gave 45% yield. Check the information. The referenced article is about chalcones.
Line 85. hydroxylethylproolidone. Do you mean. pyrrolidone?
Table 1. Why the ratio of 3a:3a’ is not determined for entries 7,9-15, 20, 22-24, 27-31?
Line 171. Table 4 is not about MBH carbonates.
Line 183. It seems there is no need to give a reference here if you describe these transformations in this article.
Lines 185-186. Check the numbers of the compounds: 7 should be 8 and 4a should be 9
Scheme 2. Transformation of 5a. This scheme is not only about 5a
Materials and Methods
Lines 210-212 were calibrated using residual undeuterated solvent as an internal reference (1H NMR= 2.50, 13C NMR= 39.52). Most of the NMR spectra in the supplementary are automatically calibrated to TMS, not DMSO. This is not a problem, since the spectra are given in the supplementary. Just remove what the spectra are calibrated to DMSO.
Check grammar in section 4.5
Line 215. Add the meaning of ψt for NMR spectra
Line 228. at rt‐reflux. Room temperature or reflux?
Line 230. EtOAc:PE. Use a hyphen instead of a colon
Line 242. ethyl Morita‐Baylis‐Hillman carbonate. Use the systematic name.
Line 768. The stereoselectivity of products was explained by the sing crystal x‐ray of 4a. The stereochemistry was confirmed for compound 4a. But it is not clear how exactly it was explained
Line 769. Further exploration and application of this reaction in organic synthesis is ongoing in our laboratory. Is this the important conclusion of this article? Maybe this sentence should be removed.
Is it possible to distinguish the signals of diastereomers (3q+3'q, 3u+3'u and the rest) in NMR spectra and describe them separately?
Check the NMR spectra. For many compounds, the number of signals does not match the number of carbon atoms. For example, for compound 3'p (C21H14BrN5O2) due to 1,4-substituted phenyl, the number of signals should be 19, not 20.
supplementary Page 15. There are notes in Chinese.

Author Response
We would like to thank Ms. Katarina Modic and the reviewers for constructive suggestions. We carefully revised this manuscript. The key revisions are listed below which have also been marked in the manuscript.
1) Line 33, Figure 1. camptothecin minics. Do you mean . camptothecin mimics?
√ We have revised and marked in the manuscript.
2) Lines 44 ...check spelling (diploarophile, phenyldenemalononitrile). See attached file (highlighted in red).
√ We have revised and marked in the manuscript.
3) Figure 1. ACC inhibitor should be III.
√ We have revised.
4) Scheme 1. Do not specify the configuration for 6' or provide the structure of another diastereomer.
√ We have added another isomer in Scheme 1.
5) LIne 59. imines become be urgent. Check grammar.
√ We have revised.
6) LIne 77. reacted smoothly with 2a and gave 45% yield. Check the information. The referenced article is about chalcones.
√ The referenced article was reported by ourselves. Initially, we performed this reaction using the optimum condition of this reference.
7) Line 85. hydroxylethylproolidone. Do you mean. pyrrolidone?
√ We have revised.
8) Table 1. Why the ratio of 3a:3a’ is not determined for entries 7,9-15, 20, 22-24, 27-31?
√ Because the yields were too low to determine the ratio of isomers for these entries.
9) Line 171. Table 4 is not about MBH carbonates.
√ We have revised.
10) Line 183. It seems there is no need to give a reference here if you describe these transformations in this article.
√ We have revised.
10) Lines 185-186. Check the numbers of the compounds: 7 should be 8 and 4a should be 9.
√ We have checked and revised.
11) Scheme 2. Transformation of 5a. This scheme is not only about 5a.
√ We have revised.
12) Lines 210-212 were calibrated using residual undeuterated solvent as an internal reference (1H NMR= 2.50, 13C NMR= 39.52). Most of the NMR spectra in the supplementary are automatically calibrated to TMS, not DMSO. This is not a problem, since the spectra are given in the supplementary. Just remove what the spectra are calibrated to DMSO.
√ We have deleted “and were calibrated using residual undeuterated solvent as an internal reference (1H NMR= 2.50, 13C NMR= 39.52).”
13) Check grammar in section 4.5.
√ We have checked and revised.
14) Line 215. Add the meaning of ψt for NMR spectra.
√ We have added.
15) Line 228. at rt‐reflux. Room temperature or reflux?
√ We have revised.
16) Line 230. EtOAc:PE. Use a hyphen instead of a colon.
√ We have revised.
17) Line 242. ethyl Morita‐Baylis‐Hillman carbonate. Use the systematic name.
√ We have revised.
18) Line 768. The stereoselectivity of products was explained by the sing crystal x‐ray of 4a. The stereochemistry was confirmed for compound 4a. But it is not clear how exactly it was explained.
√ We have revised and highlighted in the manuscirpt.
19) Line 769. Further exploration and application of this reaction in organic synthesis is ongoing in our laboratory. Is this the important conclusion of this article? Maybe this sentence should be removed.
√ We have deleted the sentence.
20) Is it possible to distinguish the signals of diastereomers (3q+3'q, 3u+3'u and the rest) in NMR spectra and describe them separately?
√ We have distinguished the signals of diastereomers (3q+3'q, 3u+3'u and the rest) in 1H NMR spectra and described them separately, but we have not distinguished the signals of diastereomers in 13C NMR.
21) Check the NMR spectra. For many compounds, the number of signals does not match the number of carbon atoms. For example, for compound 3'p (C21H14BrN5O2) due to 1,4-substituted phenyl, the number of signals should be 19, not 20.
√ We have checked and revised the NMR spectra. One or two signals were too short to display in some spectra.
22) supplementary Page 15. There are notes in Chinese.
√ We have deleted notes in Chinese.
We have revised the WHOLE manuscript carefully and tried to avoid any grammar or syntax error. In addition, we have asked several colleagues who are skilled authors of English language papers to check the English. We believe that the language is now acceptable for the review process.
The manuscript has been resubmitted to your journal. We look forward to your positive response.
Sincerely,
Guizhou Yue

Reviewer 3 Report
In this work, the authors have presented an interesting approach to access dicyclic spiropyridazine oxoindole derivatives was reported, using [3+3]-cycloaddition of N-unsubstituted isatin N,N’-cyclic azomethine imine 1,3-dipoles, generated from the condensation of isatins and pyrazolidones, with Knoevenagel adducts.
There are plenty of grammatical and spelling mistakes in the text. Kindly improve the language:
Line 22: Remove the space in NMR.
Line 35: This sentence is incomplete/unclear. Kindly improve the English. "the exploration of...." would be more suitable.
Line 39: This sentence is incomplete/unclear. Kindly improve the English.
Line 40: Correct the spelling to activation.
Line 42 and 43: Correct the spelling to cycloaddition.
Line 44: Correct the spelling to dipolarophile.
Line 50: Correct the spelling to dipole and spiropyridazine.
Line 58: Correct the spelling to dipolar.
Line 59: This sentence is incomplete/unclear. Kindly improve the English.
Line 60: Correct the spelling to presence.
Line 66: Correct the spelling to dipoles.
Line 77: Correct the spelling to presence.
Line 78: "reported in our.." and "Inspired from the above.."
Line 83: "gave slightly improved..."
Line 93: Correct the spelling to additionally.
Line 95: "According to the above experiments.."
Line 100: "all the yields..".
Line 101: "while the..."
Line 115: This sentence is incomplete/unclear. Kindly improve the English.
Line 120: This sentence is incomplete/unclear. Kindly improve the English.
Line 125: This sentence is incomplete/unclear. Kindly improve the English.
Line 170: This sentence is incomplete/unclear. Kindly improve the English.
Line 174: "In most reactions.."
Line 184: Correct the spelling to product.
Author Response
We would like to thank Ms. Katarina Modic and the reviewers for constructive suggestions. We carefully revised this manuscript. The key revisions are listed below which have also been marked in the manuscript.
1) Line 22: Remove the space in NMR.
√ We have revised.
2) Line 35: This sentence is incomplete/unclear. Kindly improve the English. "the exploration of...." would be more suitable.
√ We have revised.
3) Line 39: This sentence is incomplete/unclear. Kindly improve the English.
√ We have revised.
4) Line 40: Correct the spelling to activation.
√ We have revised.
5) Line 42 and 43: Correct the spelling to cycloaddition.
√ We have revised.
6) Line 44: Correct the spelling to dipolarophile.
√ We have revised.
7) Line 50: Correct the spelling to dipole and spiropyridazine.
√ We have revised.
8) Line 58: Correct the spelling to dipolar.
√ We have revised.
9) Line 59: This sentence is incomplete/unclear. Kindly improve the English.
√ We have revised.
10) Line 60: Correct the spelling to presence.
√ We have revised.
11) Line 66: Correct the spelling to dipoles.
√ We have revised.
12) Line 77: Correct the spelling to presence.
√ We have revised.
13) Line 78: "reported in our.." and "Inspired from the above.."
√ We have revised.
14) Line 83: "gave slightly improved..."
√ We have revised.
15) Line 93: Correct the spelling to additionally.
√ We have revised.
16) Line 95: "According to the above experiments.."
√ We have revised.
17) Line 100: "all the yields..".
√ We have revised.
18) Line 101: "while the..."
√ We have revised.
19) Line 115: This sentence is incomplete/unclear. Kindly improve the English.
√ We have revised.
20) Line 120: This sentence is incomplete/unclear. Kindly improve the English.
√ We have revised.
21) Line 125: This sentence is incomplete/unclear. Kindly improve the English.
√ We have revised.
22) Line 170: This sentence is incomplete/unclear. Kindly improve the English.
√ We have revised.
23) Line 174: "In most reactions.."
√ We have revised.
24) Line 184: Correct the spelling to product. This figure 1 and 2 has not been referred to within the text of the manuscript. Please add call out.
√ We have revised.
We have revised the WHOLE manuscript carefully and tried to avoid any grammar or syntax error. In addition, we have asked several colleagues who are skilled authors of English language papers to check the English. We believe that the language is now acceptable for the review process.
The manuscript has been resubmitted to your journal. We look forward to your positive response.
Sincerely,
Guizhou Yue

Round 2
Reviewer 2 Report
The manuscript has been greatly improved and may be published. There are two remarks.
Line 39: 1.3-diploar should be 1,3-dipolar
Lines 75-77: Make it clear that you mean the conditions described in this article [ref. 42] but not the reaction itself.